# Systematic analysis and prediction of genes associated with monogenic disorders on human chromosome X

Elsa Leitão [1,29], Christopher Schröder[1,29], Ilaria Parenti [1], Carine Dalle[2], Agnès Rastetter[2], Theresa Kühnel [1], Alma Kuechler[1], Sabine Kaya[1], Bénédicte Gérard[3], Elise Schaefer[4], Caroline Nava [2], Nathalie Drouot[5,6,7,8], Camille Engel[5,6,7,8], Juliette Piard [9,10], Bénédicte Duban-Bedu[11], Laurent Villard [12,13], Alexander P. A. Stegmann [14,15], Els K. Vanhoutte[15], Job A. J. Verdonschot [15,16], Frank J. Kaiser[1], Frédéric Tran Mau-Them [10,17], Marcello Scala[18,19], Pasquale Striano [18,19], Suzanna G. M. Frints[15,20], Emanuela Argilli[21,22], Elliott H. Sherr[21,22], Fikret Elder [23], Julien Buratti [23], Boris Keren[23], Cyril Mignot[2,24], Delphine Héron[24], Jean-Louis Mandel[3,5,6,7,8], Jozef Gecz [25,26,27], Vera M. Kalscheuer [28], Bernhard Horsthemke [1], Amélie Piton [3,5,6,7,8] & Christel Depienne [1] ✉

Disease gene discovery on chromosome (chr) X is challenging owing to its unique modes of inheritance. We undertook a systematic analysis of human chrX genes. We observe a higher proportion of disorder-associated genes and an enrichment of genes involved in cognition, language, and seizures on chrX compared to autosomes. We analyze gene constraints, exon and promoter conservation, expression, and paralogues, and report 127 genes sharing one or more attributes with known chrX disorder genes. Using machine learning classifiers trained to distinguish disease-associated from dispensable genes, we classify 247 genes, including 115 of the 127, as having high probability of being disease-associated. We provide evidence of an excess of variants in predicted genes in existing databases. Finally, we report damaging variants in *CDK16* and *TRPC5* in patients with intellectual disability or autism spectrum disorders. This study predicts large-scale gene-disease associations that could be used for prioritization of X-linked pathogenic variants.

Sex in mammals is determined by a diverging pair of sex chromosomes (chr). Human females have two copies of the 156-Mb chrX while males have a single X copy and a smaller 57-Mb chrY. Compensation of gene dosage in females is achieved through X chromosome inactivation (XCI), a process leading to the epigenetic silencing of an entire chrX, apart from two pseudoautosomal regions (PARs). This process happens during early embryonic development, is randomly and independently established in each cell, and stably maintained during further cell divisions[1,2]. As a consequence, female individuals are cell mosaics,

each cell expressing genes from either the maternal or paternal X copy[3]. A subset of genes, which can be variable between individuals and tissues, escapes X inactivation and continues to be expressed from both X chromosomes[4].

The last two decades have revolutionized concepts of X-linked inheritance, by depicting its unique but multiple forms[1,5]. The first and most widely described disorders (>100 genes) mainly affect hemizygous males and are transmitted through healthy or mildly symptomatic female carriers. Other modes of X-linked inheritance are mainly

observed in disorders affecting the central nervous system. Variants in X-linked genes such as *MECP2*[6], *CDKL5*[7,8], and *DDX3X*[9] preferentially affect heterozygous females. These variants usually occur de novo, and hemizygous males are either not viable or survive only if variants are hypomorphic or mosaic. Variants in other genes affect hemizygous males and heterozygous females almost equally. The list of X-linked disorders first described as selectively affecting males but turning out to also affect females is continually increasing and include *IQSEC2*[10], *NEXMIF*[11], *KDMSC*[12], *HUWE1*[13], *USP9X*[14], and *CLCN4*[15]. Lastly, two X-linked disorders, related to *PCDH19* and *EFNB1*, affect heterozygous females and postzygotic somatic mosaic males (due to cellular interference), while hemizygous males are spared[16,17].

Disease gene discovery on chrX is thus associated with greater challenges, including male-female patient selection and variant interpretation biases, compared to autosomes. ChrX is often omitted from genome-wide analyses in a research context due to the difficulty of dealing with sex dichotomy in bioinformatics pipelines. Identifying novel gene-disease associations on chromosome X requires dedicated studies of families with multiple affected males[18,19] or multiple subjects with matching phenotypes[9]. The interpretation of X-linked variants in sporadic cases and small families remains difficult in absence of extended segregation in the family, which is rarely available. Furthermore, the presence of damaging variants at relatively high frequency in large databases (ExAC, gnomAD) led to question previously established gene-disease associations[20]. The Deciphering Developmental Disorders (DDD) consortium recently estimated that X-linked disorders overall affect males and females equally and represent 6% of developmental disorders[21]. However, despite the large size of the cohort (11,044 affected individuals), this study failed to identify new X-linked disorder-associated genes.

In this work, we first undertook a systematic analysis of all coding genes on human chromosome X and compared the proportion and characteristics of associated disorders to those on autosomes. In a second step, we investigated the relevance of multiple variables to predict gene-disorder associations. Lastly, we used these predictions to uncover new disease-gene associations supported by the literature as well as by patient data and functional studies.

## Results

### Chromosome X is enriched in disorder genes and in genes relevant to brain function

Chromosome X comprises 829 protein-coding genes annotated in HUGO Gene Nomenclature Committee (HGNC), including 205 associated with at least one monogenic disorder (referred to as 'disorder genes') in OMIM (Fig. 1a; Supplementary Data 1). We used the clinical synopsis to compare the proportion of disorder genes and their associated clinical features on chrX (available for 202 genes) and autosomes (Methods, Fig. 1b). We observed a significant and specific enrichment in disorder genes on chrX (24% versus 12-22%, $p = 1.87 \times 10^{-3}$; OR = 1.5; Fig. 1c, d and Supplementary Data 2-4). Furthermore, genes on chrX were significantly more frequently associated with neurological phenotypes than genes on autosomes (77% versus 55-76%; $p = 7.01 \times 10^{-3}$; OR = 2.0; Fig. 1e, f). More specifically, we observed that chrX is enriched in genes associated with intellectual disability (ID; 58% versus 27-45%; $p = 5.92 \times 10^{-11}$; OR = 2.9), seizures (46% versus 23-38%; $p = 1.12 \times 10^{-4}$; OR = 2.1) and language impairment (32% versus 11-24%; $p = 5.59 \times 10^{-3}$; OR = 2.0; Fig. 1g–l), but not motor development, spasticity, or ataxia (Fig. 1m, n; Supplementary Fig. 1a, b). The difference remained significant when genes associated with provisional gene-phenotype relationships (P), susceptibility to multifactorial disorders (M), or traits (T) (referred to as 'PMT' genes) were included in the comparison (Supplementary Fig. 1c, d; Supplementary Data 1). In total, 84% of known disease genes on chrX are associated with ID, seizures, or language impairment and about 30% are associated with all three clinical outcomes (Fig. 2).

### Confirmed disorder-associated genes share specific features

Despite chrX being enriched in disorder-associated genes, 598 genes (71%) are not yet related to any clinical phenotype (referred to as 'no-disorder genes'; Fig. 1a; Supplementary Data 1). We hypothesize that disorder genes share specific common features that dispensable genes do not exhibit and that could be used to predict genes that remain to be associated with human disorders. To test this hypothesis, we retrieved annotations from different sources and/or calculated additional metrics, including: (1) gnomAD gene constraint metrics: LOEUF (rank of intolerance to loss-of-function (LoF) variants), misZ (intolerance to missense variants score) and synZ (intolerance to synonymous variants score); (2) coding-sequence (CDS) length; (3) the degree of exons and promoter conservation across 100 species; (4) the promoter CpG density;[22] and (5) gene expression data, conveyed as a tissue specificity measure (tau) and brain-related expression levels from GTEx and BrainSpan resources (Fig. 3a; Supplementary Data 2).

We investigated whether the distribution of these variables differ between disorder genes and no-disorder genes. As expected, disorder genes had lower LOEUF values and higher misZ scores but similar synZ scores than no-disorder genes. We observed a significant enrichment of disorder genes in the three lowest LOEUF deciles and in the two highest misZ deciles, with the distributions of LOEUF and misZ metrics in the PMT group showing intermediate values between disorder and no-disorder genes (Fig. 3b, c). Disorder genes were enriched in the highest decile of exon conservation, promoter conservation scores and CDS length distribution with an overall distribution that was variable in all groups (Fig. 3d; Supplementary Fig. 2). Disorder genes tended to present higher promoter CpG density than no-disorder genes, although the difference was not significant. Remarkably, we observed that disorder genes are more broadly expressed (tau < 0.6) and show intermediary levels of gene expression in brain-tissues compared to no-disorder genes, for which expression was more often restricted to a few tissues (tau > 0.6) and are less expressed in the brain (Supplementary Fig. 2).

We next investigated how the existence of close paralogues and the location in PARs may influence their relationship to human disease. From the 18 PAR protein-coding genes, only two (*SHOX* and *CSF2RA*) have been associated with a medical condition so far. We also observed that disorder genes are significantly depleted in close paralogues compared to PMT/no-disorder genes ($p = 3.1 \times 10^{-6}$; OR = 0.33; Supplementary Fig. 3a, b). This suggests that highly similar paralogues might compensate pathogenic variants in corresponding X-linked genes.

Previous comparisons suggest that LOEUF, misZ, and exon conservation scores, which showed marked differences between the two subgroups, can best differentiate disorder from no-disorder genes. We then focused on the deciles enriched in disorder genes to list the genes in the no-disorder group exhibiting similar characteristics (Fig. 3a). More specifically, we used the limits of those deciles to define thresholds for LOEUF, misZ, and exon conservation scores as follows: LOEUF ≤ 0.326 (L), misZ ≥ 2.16 (M), and/or exon-score ≥ 0.9491 (E). 149 of the 205 (73%) disorder genes met one or more L, M or E (LME) criteria (Fig. 3e). Among the 205 disorder genes, 56 genes failing to meet any of the LOEUF/misZ criteria had a smaller CDS length ($p = 4.20 \times 10^{-9}$), indicating that the performance of LOEUF and misZ in differentiating disorder from no-disorder genes depends on CDS length (Supplementary Fig. 3c). Exon conservation alone allowed retrieving eight disorder genes with small CDS for which LOEUF and misZ failed to reach the thresholds (Supplementary Fig. 3c; Supplementary Data 2). When applying the same thresholds to the 598 genes not associated with disease, 127 (21%) genes fulfilled at least one condition and 35 fulfilled at least two criteria (Fig. 3e; Supplementary Data 2). Half ($n = 13$) of the PMT genes also met at least one criterion and seven at least two criteria (Fig. 3e). Altogether, 140 genes shared at least one of the LME criteria with known disorder genes.

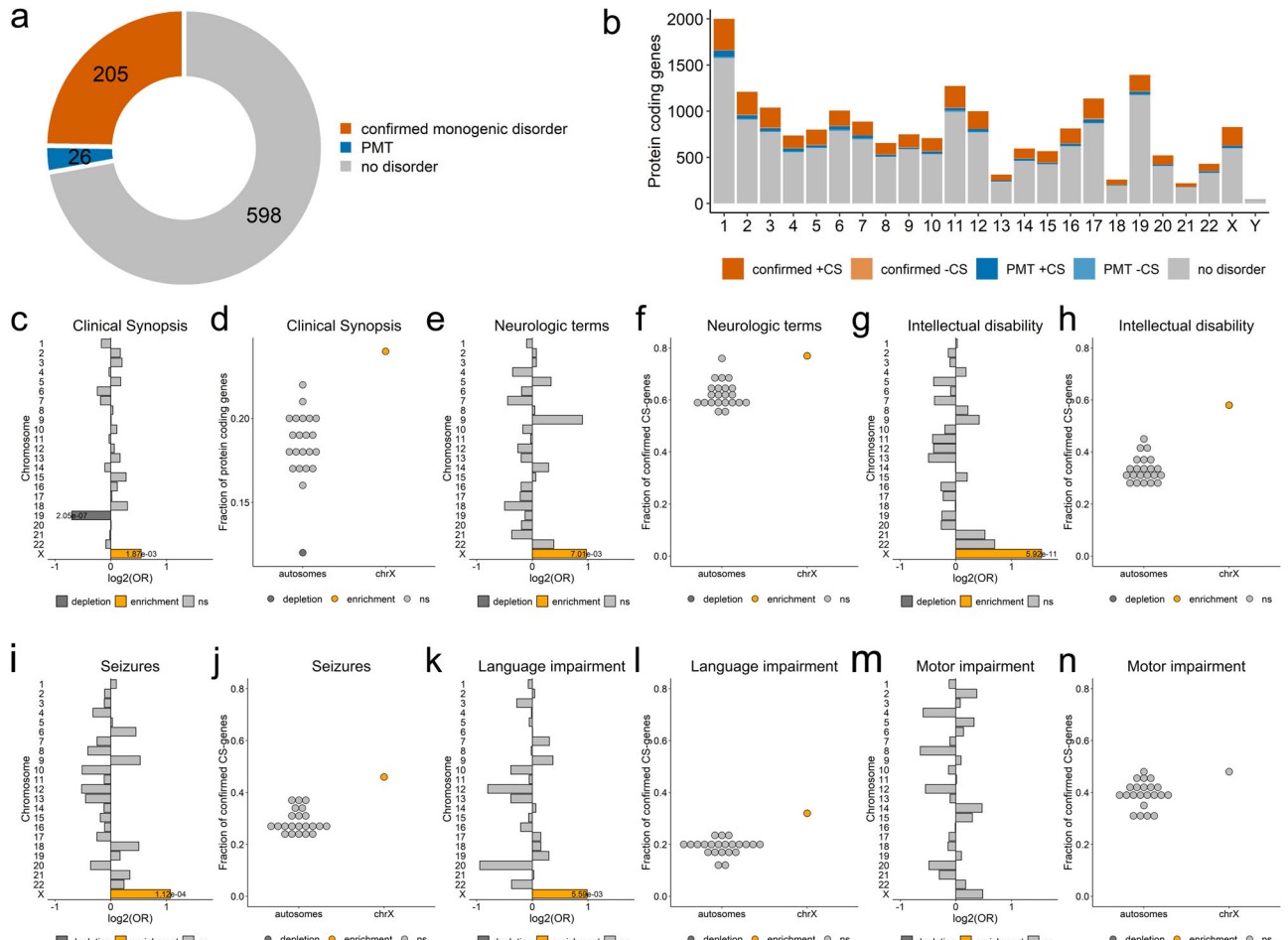

**Fig. 1 | Disorder genes across chromosomes. a** Number of protein-coding genes on chrX. Genes associated with at least one monogenic disorder: orange, confirmed genes; blue: genes with provisional associations (P), associated with susceptibility factors to multifactorial disorders (M) or traits (T) (altogether: PMTs); grey: genes without known phenotypes (no-disorder genes). **b** Number of protein-coding genes per chromosome. Dark and light colors represent genes with Clinical Synopsis (CS) data for at least one of the associated phenotypes (+CS) or without CS (-CS), respectively. **c** Per chromosome enrichment/depletion of protein-coding genes with at least one associated phenotype comprising Clinical Synopsis data (confirmed CS-genes). **d** Fraction of confirmed CS-genes among protein-coding genes. **e–n.** Predominance of chrX genes associated with neurologic features. **e, g,**

**i, k and m,** Per chromosome enrichment/depletion of genes with non-specific neurologic features (**e**), intellectual disability (**g**), seizures (**i**), language impairment (**k**), or motor development (**m**). **f, h, j, l and n,** Fraction of genes associated with non-specific neurologic features (**f**), intellectual disability (**h**), seizures (**j**), language impairment (**l**), or motor development (**n**). **c–n,** Yellow, enrichment; dark-grey, depletion; light grey, not significant (ns). Terms corresponding to the same neurological clinical features were used in OMIM searches (Supplementary Data 3). Only significant *p*-values are shown (Fisher's test (two-sided) followed by Bonferroni correction for multiple testing across both chromosomes and phenotypes). **a–n** Source data are provided as a Source Data file.

## Prediction of novel disorder-associated genes on chrX

The threshold approach was limited by the number of variables that could be taken into account to differentiate disorder from no-disorder genes. We then used machine learning to predict remaining disorder genes in a more systematic and unbiased fashion (Fig. 4a). As most X-linked disorders are associated with neurological features, we decided to distinguish two classes of genes: (1) known disorder genes associated with neurological features without homozygous LoF in gnomAD (Cbi genes; *n* = 2,170), and (2) dispensable genes not associated with any known disorders (NDt genes; *n* = 1456), with dispensable referring to genes with at least one homozygous LoF variant in gnomAD, as defined by Karczewski et al.[23] (Methods; Fig. 4b; Supplementary Data 1). To this aim, we collected 83 variables from genes on all chromosomes (*n* = 19,154), including 35 gene constraints features from gnomAD[23], two nucleotide conservation metrics, 35 features related to expression data stratified according to sex (19 based on GTEx multi-tissue data from adults[24] and 16 based on Brainspan data focusing on brain development[25]), gene structure attributes (*n* = 4), relative position of the gene on the chromosome (*n* = 2), and

data on paralogues (*n* = 5) (Methods; Supplementary Data 5 and 6). We then trained and compared 25 different machine learning models provided by scikit-learn[26] to distinguish the abovementioned Cbi and NDt gene classes (Methods; Fig. 4b; Supplementary Data 1, 5 and 6). We applied nested cross-validation and used Matthews correlation coefficient (MCC) to evaluate the performance of each classifier (Supplementary Fig. 4). We selected the top five performing classifiers, AdaBoostClassifier, BaggingClassifier, LinearSVC, MLPClassifier, and RandomForestClassifier (ML-classifiers; Supplementary Data 7) for further analysis. ML-classifiers used most input features for prediction although the importance of these features differ between classifiers (Supplementary Fig. 5). Altogether, the five best ML-classifiers exhibited a median sensitivity of 95% for known disorder genes on chromosome X (FDR < 0.05 for each), while sensitivity on autosomes was somewhat lower (range of medians on autosomes: 63–81%; Supplementary Fig. 6; Supplementary Fig. 7). We integrated the predictions of the ML-classifiers and considered genes with (i) probability > 0.5 in the five classifiers or (ii) a mean global probability > 0.82 (FDR < 0.05). Out of the 598 no-disorder genes on chrX, each classifier predicted

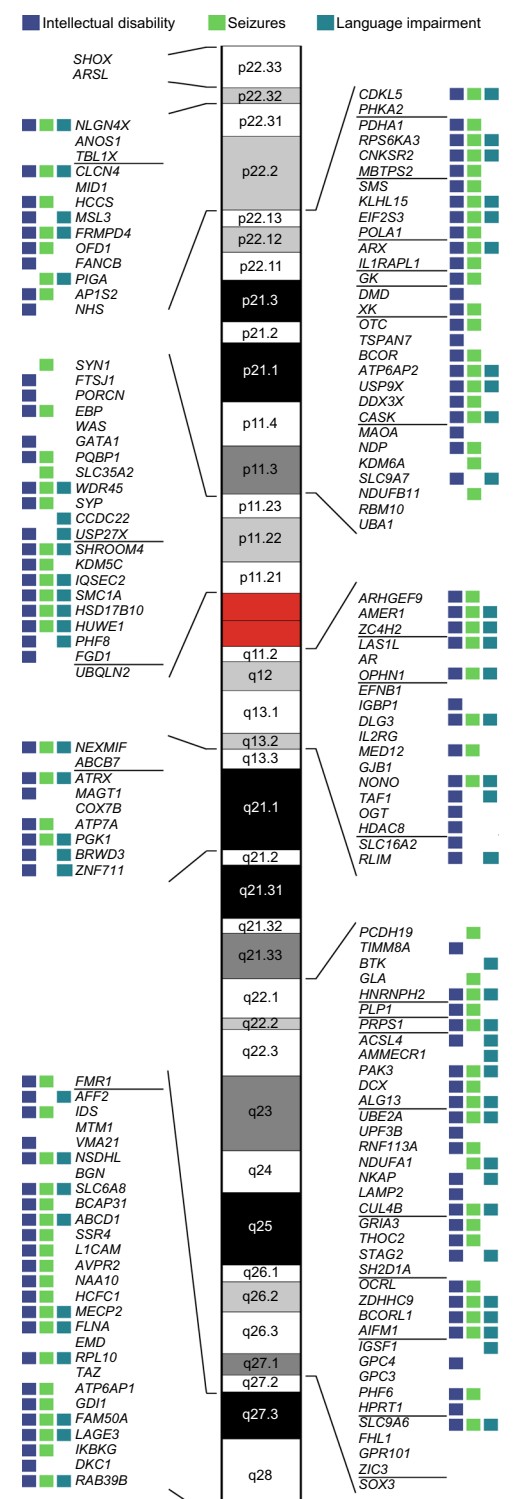

**Fig. 2 | Scheme of genes associated with neurological features on chrX.** Squares next to the genes represent association with intellectual disability (blue), seizures (green) or language impairment (cyan). A horizontal line separates genes present in different chromosome bands. Data underlying this scheme can be found in Supplementary Data 2. The banded chrX was generated using karyoploteR[85].

established LME thresholds (Fig. 4d). The ML-approach corroborated 20 of the 26 PMT genes as disease-associated, 13 (65%) of which also met LME criteria (Fig. 4d; Fig. 5a, b). From the 247 no-disorder genes on chrX predicted by the ML-classifiers, 115 (47%) also met at least one of the LME criteria (Fig. 4d), with four meeting all three, 29 two and 82 genes one criterion (Fig. 5c). The 132 remaining genes predicted by the ML-classifiers did not meet any LME criteria (Fig. 4d; Fig. 5d), whereas only 12 meeting LME criteria were not predicted by all five ML-classifiers using stringent conditions (mean global probability > 0.82; Fig. 4d; Fig. 5e). However, 11 out of these 12 genes were predicted (probability > 0.5) by at least four of the ML-classifiers (Fig. 5e).

The ML-classifiers were able to capture the relevance of features other than LOEUF, misZ, and exon conservation to uncover a large fraction of putative disease-associated genes. We evaluated whether the distribution of LOEUF, misZ, and exon conservation, as well as the top five most important features for each ML-classifier (Supplementary Fig. 5), differ between no-disorder non-predicted genes and known/predicted disease-associated genes. Along with LOEUF, the features max_af, classic_caf, lof_z, pLI, and pNull, which all relate to loss-of-function intolerance constraints, show markedly different distributions between non-predicted and known/predicted disease-associated genes (Supplementary Fig. 8a–f; Supplementary Data 5). Likewise, mis_z and oe_mis_pphen relate to missense variant constraints and their distribution is similar for predicted and known disorder genes (Supplementary Fig. 8g-h; Supplementary Data 5). Exon_score conservation, as well as gene expression metrics (mean_Pre_M, var_Pre_M, tau_1, and tau_2), show highly overlapping distributions between predicted and known disorder genes (Supplementary Fig. 8i-m; Supplementary Data 5).

We also observed that, similarly to disorder genes, ML-predicted genes are significantly depleted in close paralogues compared to non-predicted no-disorder genes ($p = 5.6 \times 10^{-15}$; OR = 0.24; Supplementary Fig. 8n,o). This suggests that abolished functions of predicted genes cannot be compensated by any paralogue.

Altogether, these results suggest that our ML approach is a valid method to predict genes remaining to be associated with disease with high accuracy.

### Validation of putative disorder genes

To validate our predictions, we searched for evidence from the literature or existing databases that variants in the 247 ML-predicted genes could result in new X-linked disorders (Fig. 6a). We first used HGMD and DECIPHER to retrieve the number of single nucleotide variants and small indels of unknown significance reported in each gene (see Methods) and compared the number of variants in predicted versus non-predicted gene categories (Supplementary Data 8). This analysis revealed that the 247 ML-predicted genes are enriched for genes with reported variants in HGMD and/or Decipher compared to non-predicted genes ($p = 2.7 \times 10^{-5}$; OR = 2.1). ML-predicted genes had on average more point variants than non-predicted genes (Fig. 6b,c; Supplementary Fig. 9), suggesting that this excess is due to pathogenic variants.

Second, we used the expert curated database of gene-disease relationships in neurodevelopmental disorders (SysNDD)[27] and the provided curations from Gene2Phenotype, PanelApp, Radboudumc, SFARI, Geisingel_DBD, OMIM_NDD, and Orphanet_id to compare the overlap with predicted and non-predicted genes (Fig. 6a). Fifteen of the 20 ML-predicted PMT genes were present in at least one of the gene databases (Supplementary Fig. 10). Three genes (*PTCHD1*, *NLGN3*, and *KIF4A*), indicated as definitive SysNDD genes, were present in multiple databases. *PTCHD1* and *NLGN3* are associated with susceptibility to autism in OMIM, but both were recently confirmed to cause a monogenic ID disorder frequently associated with autism[28–30]. A splice site variant in *KIF4A* was identified in a family with four affected males[31], and this finding was recently strengthened by the identification of

between 323 and 393 (probability > 0.5), 287 of which were shared by all five classifiers (Fig. 4a, c). Of these, 247 genes were predicted as putative brain disorder-associated with a FDR < 0.05 (ML-classifiers mean probability) (Fig. 4a, b; Supplementary Data 6). A large proportion of ML-predicted confirmed genes (144/189; 76%) met previously

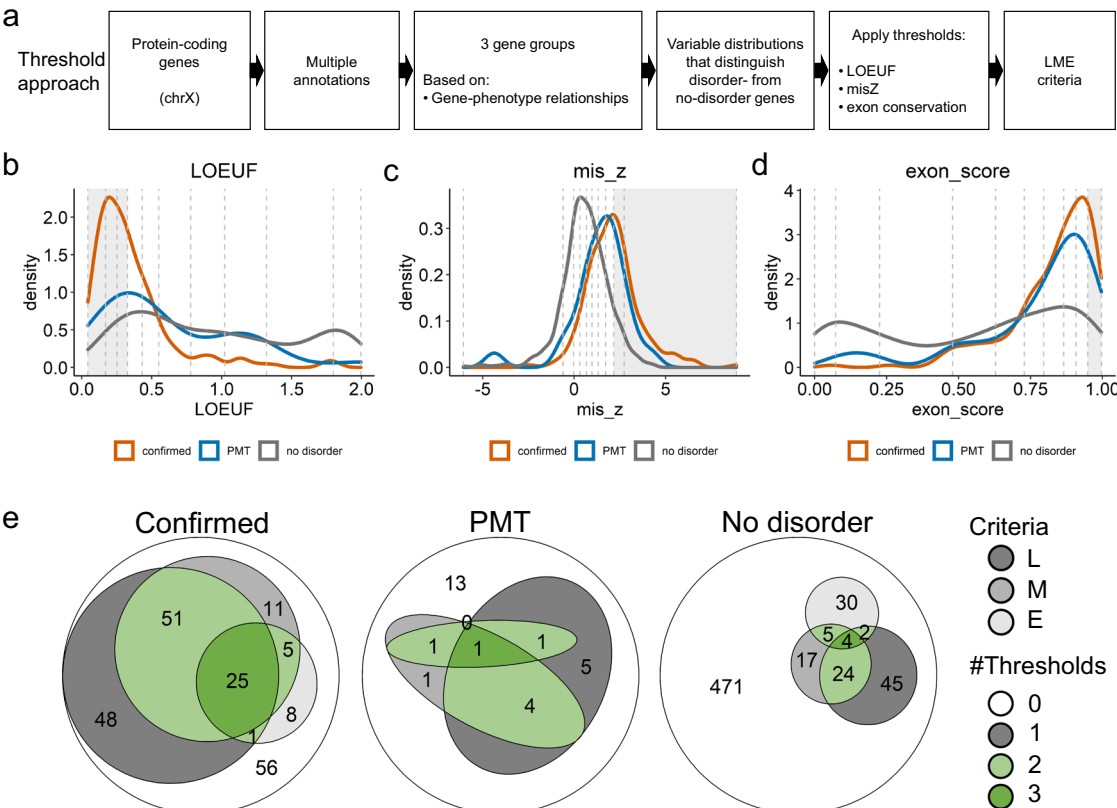

**Fig. 3 | Features shared by known disorder genes. a** Overview of the approach used to uncover features shared by disorder genes (details in Methods). Density plots showing the distribution of LOEUF (**b**), misZ (**c**) and exon-conservation score (**d**) according to gene group. Confirmed disorder genes (orange), PMT genes (blue), no-disorder genes (grey). Vertical dashed lines separate deciles of the overall distribution. Grey areas depict deciles for which confirmed disorder-associated genes are enriched (related to Supplementary Fig. 2). **e** Euler diagrams showing the

number of genes fulfilling LOEUF (L), misZ (M) and exon-conservation (E) criteria for confirmed (left), PMT (middle) and no-disorder genes (right). Thresholds: LOEUF ≤ 0.326, misZ ≥ 2.16, exon-conservation score ≥ 0.9491. Genes fulfilling at least two LME criteria are shown in green (light or dark green for two or three criteria, respectively). Genes meeting only one of the metrics are shown in different shades of grey (dark to light: LOEUF, misZ, exon-conservation). Genes not meeting LME criteria are shown in white. **b**–**e**, Source data are provided as a Source Data file.

additional de novo or inherited variants causing a range of different phenotypic manifestations[32,33]. Focusing on no-disorder genes, 70 (28%) of genes predicted by ML and 69 (20%) of the non-predicted genes were present in at least one database, with predicted genes being present in significantly more databases than non-predicted genes (p = 0.02; OR = 2.2; Fig. 6d, e). Four predicted genes (*FGF13*, *PLXNA3*, *OTUD5* and *GLRA2*) were considered definitive NDD genes in SysNDD database, while 16 others were considered as having limited evidence (Fig. 6e). Pathogenic variants in *OTUD5* and *FGF13* respectively cause a severe neurodevelopmental disorder with multiple congenital anomalies and early lethality[34,35], and developmental and epileptic encephalopathy[36], both described in 2021. Rare intragenic deletions or missense variants in *GLRA2* were first reported to be associated with autism spectrum disorders (ASD)[37] in 2016 but independent confirmation was only published in 2022[38,39]. Furthermore, variants in this gene have also recently been associated with high myopia[40]. Similarly, variants in *PLXNA3* have been associated with ID and ASD[41], or hypogonadotropic hypogonadism[42], although this gene-disease association does not yet appear in OMIM. Fifteen genes present in SysNDD but not predicted by ML showed limited evidence as NDD genes. However, one of them, *GABRA3*, is part of the 12 genes fulfilling LME criteria, and is predicted with probability > 0.5 by all five classifiers and with FDR < 0.05 by RandomForestClassifier (Fig. 5e; Supplementary Data 6 and 8). Additional evidence from the literature not yet reflected in OMIM or any other database include possibly pathogenic variants described in one or a few families in *ZMYM3*[43], *GPKOW*[44], and *WNK3*. Loss-of-function and missense variants in *WNK3*

have been identified in six different families with intellectual disability and variable epilepsy in June 2022 during the revision of this article[45].

Third, we retrieved the number of de novo predicted damaging (truncating or CADD ≥ 25) variants identified in these genes from the Martin et al.[21] and Kaplanis et al.[46] studies. In parallel, we examined exome data from two additional cohorts of patients with developmental disorders, containing mainly patients with intellectual disability (6500 from 2346 families from the Paris-APHP cohort and 1,399 individuals from 463 families from the UCSF cohort) and extracted predicted damaging variants in LME and ML-predicted genes (Supplementary Data 9). This led us to select 13 genes containing de novo damaging variants and additional predicted damaging variants identified in independent cohorts (Fig. 7). These genes include *GLRA2* and *GABRA3*, as well as 11 genes for which evidence remained limited so far. Interestingly, 19 out the 55 missense variants (34.5%) reported in these 13 genes are located in known functional protein domains (Fig. 7; Supplementary Data 10). This proportion is not however significantly higher than that expected by chance (p = 0.0549, one-tailed binomial test; considering that 2683 out of 11,089 amino acids composing the 13 proteins, i.e. 24%, are located within domains). Among those genes, *SMARCA1* has been associated with syndromic intellectual disability and Coffin-Siris-like features by the Clinical Genome Resource (ClinGen) with moderate evidence[47]. Altogether, data extracted from both databases and the literature provide additional support for gene-disease association for 20 out of the 247 predicted genes. The features underlying ML predictions obtained for each of these genes are depicted in Supplementary Fig. 11.

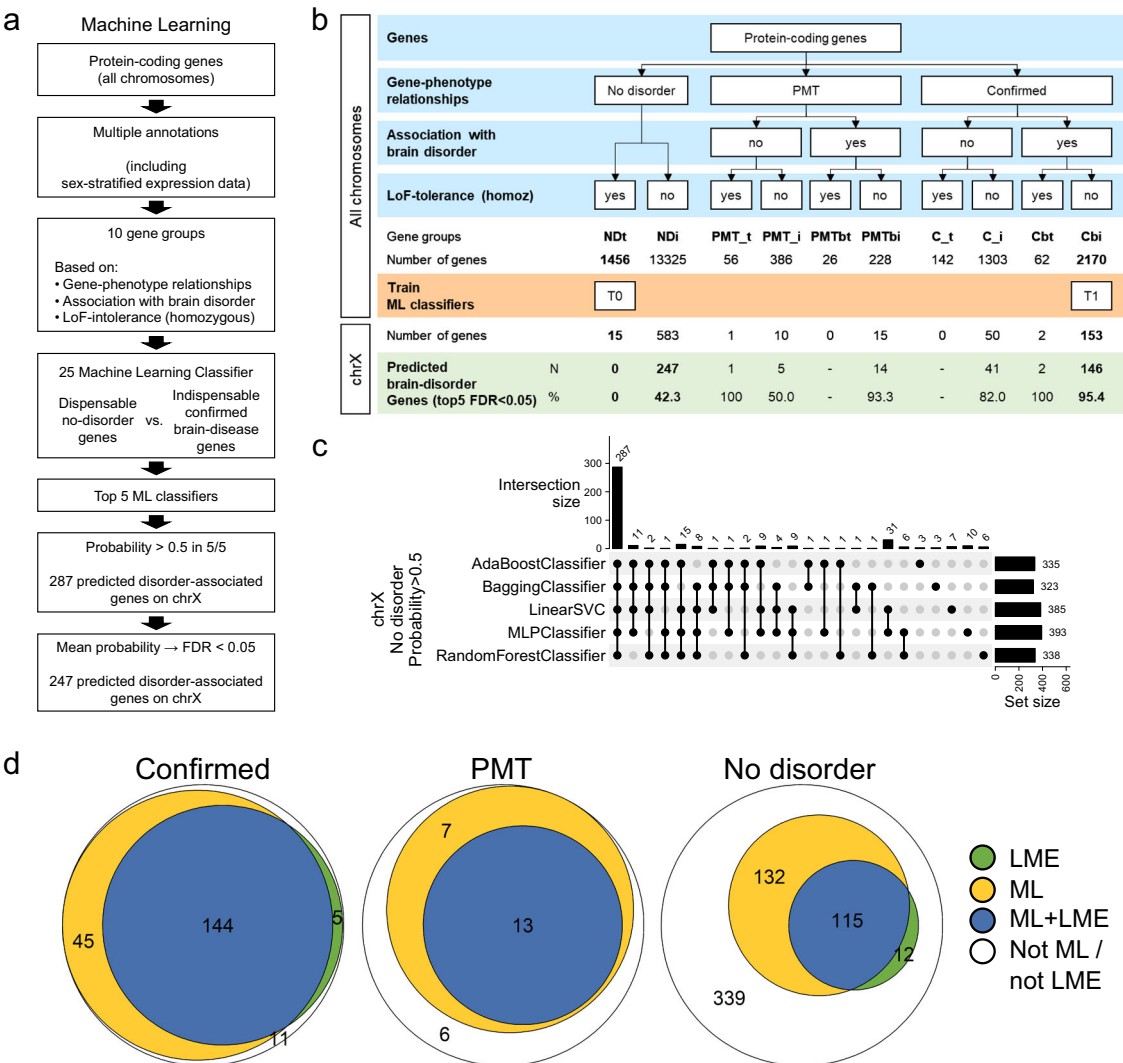

**Fig. 4 | Machine learning predicts putative disorder genes. a** Overview of the machine learning approach used to predict disorder genes (details in Methods). **b** Protein-coding genes were pre-classified into 10 subgroups based on (1) the type of associations with disorders and/or traits (confirmed, PMT, no-disorder), (2) the association with a brain disorder, and (3) the tolerance to loss-of-function (LoF) homozygous variants. Two classes were used to train the 25 machine learning classifiers: Cbi (confirmed brain-disorder associated genes that are LoF-homozygous intolerant; value 1.0) and NDt (no-disorder genes tolerant to LoF-homozygous mutations; value 0). We show number and fraction of predicted genes for each of the 10 classes for chrX (FDR < 0.05 of the mean probability of the top

five ML classifiers: AdaBoostClassifier, BaggingClassifier, LinearSVC, MLPClassifier, and RandomForestClassifier). Data underlying this scheme can be found in Supplementary Data 6. **c** Upset plot showing the number of genes predicted by each of the top five ML classifiers (set size) and the number of genes shared between classifiers (intersection size). **d** Euler diagrams showing the number of genes fulfilling LME criteria (LME, green), predicted by the ML approach (yellow) or both (blue) for confirmed (left), PMT (middle) and no-disorder genes (right). Genes not predicted by the ML approach are shown in white. **c, d** Source data are provided as a Source Data file.

## Variants in *CDK16* and *TRPC5* in patients with intellectual disability

Lastly, we focused on *CDK16* and *TRPC5*, both predicted by the ML and meeting LME thresholds but for which genetic evidence was limited (Fig. 8a, b). For these two genes, we used Genematcher[48] and identified at least one family supporting X-linked inheritance and additional variants in patients with similar phenotypes.

*CDK16* encodes a protein kinase involved in neurite outgrowth, vesicle trafficking, and cell proliferation[49]. A deletion of two nucleotides leading to a frameshift (NM_006201.5: c.976_977del, p.(Trp326Valfs*5)) segregating in four males with ID, ASD, absence seizures, and mild spasticity was reported in this gene by Hu et al.[18] Using exome sequencing, we identified a nonsense variant (c.961 G > T, p.(Glu321*)) in a 42-year-old patient with ID and spasticity. A missense variant (c.1039G > T, p.(Gly347Cys)) affecting a highly conserved amino acid of the kinase domain (CADD PHRED score: 32) was identified by genome

sequencing in a male patient with ID, ASD, and epilepsy, whose family history was compatible with X-linked inheritance (Fig. 8a, c; Supplementary Data 11). In addition, a nonsense variant (c.46C > T, p.(Arg16*)) was recently reported in a patient with ASD by Satterstrom et al.[50].

*TRPC5* encodes the short transient receptor potential channel 5, a channel permeable to calcium predominantly expressed in the brain[51]. We identified a missense variant in this gene (NM_012471.2:c.523C > T, p.(Arg175Cys), CADD PHRED score: 29.8) in three brothers with ID and ASD. The variant was inherited from the asymptomatic mother and was absent from the maternal grandparents (Fig. 8b, d; Supplementary Data 11). We investigated the basal properties of TRPC5 p.(Arg175Cys) mutant channel using whole-cell patch-clamp. Immediately after break-in, HEK293 cells expressing mutant TRPC5 exhibited an increase in immediate current recorded compared with cell expressing wild-type TRPC5 ($p = 0.003$ for inward current and $p = 0.001$ for outward current; $n = 19$ for WT, $n = 22$ for mutant; Kruskal-Wallis one-way

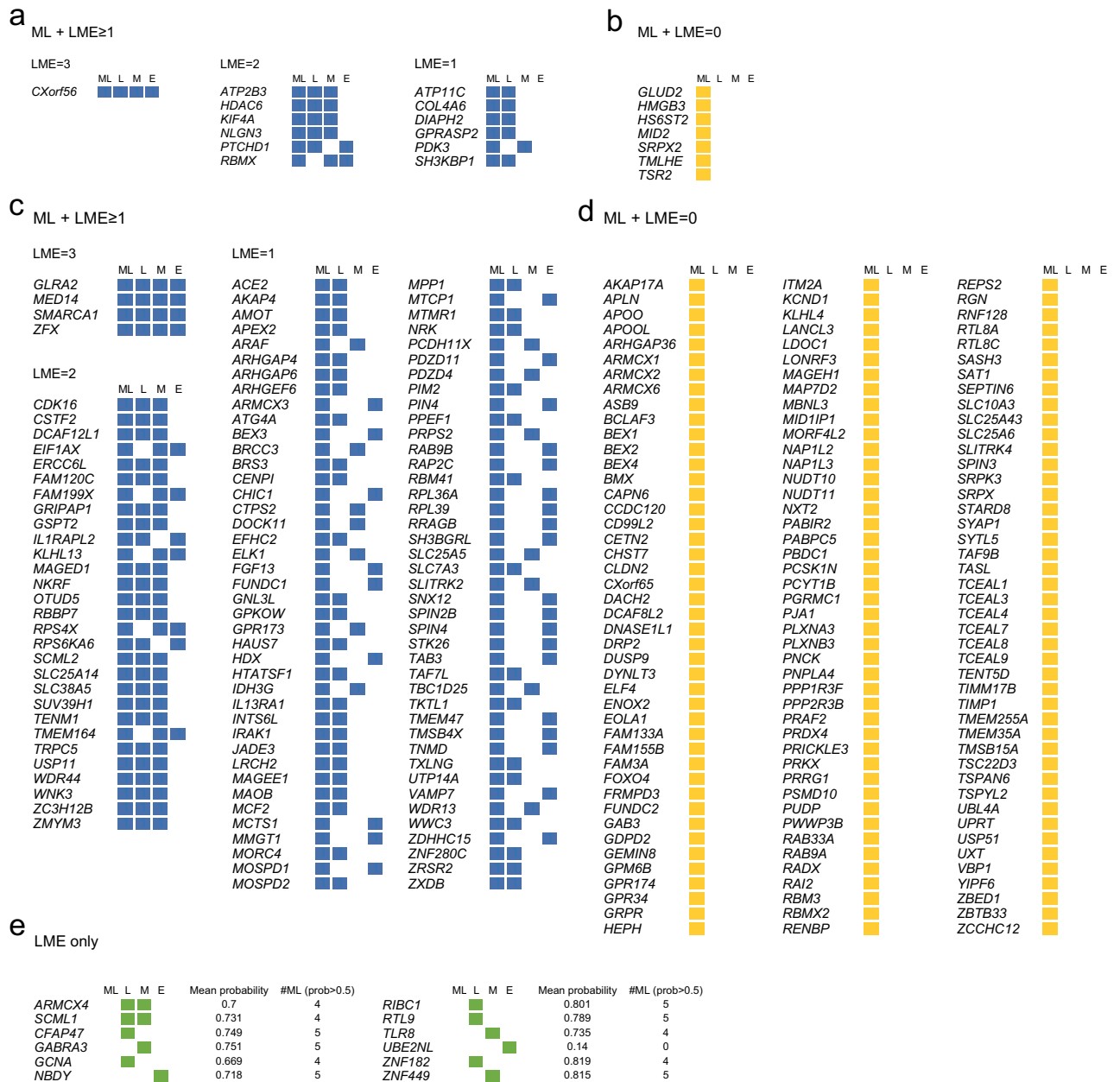

**Fig. 5 | Different evidence strength for the predicted disorder-associated genes.** PMT (**a**, **b**) and no-disorder genes (**c**, **d**) predicted by the ML-classifiers as possibly disease-associated, either fulfilling (**a**, **c**) or not LME criteria (**b**, **d**). **e** No-disorder genes meeting LME criteria but not predicted by the ML-classifiers. Values show for each gene the mean probability of the five ML-classifiers and the number of ML-classifiers showing probability > 0.5. **a–e** Blue, genes predicted by the ML approach and fulfilling at least one LME criteria; yellow, genes predicted by the ML approach; green, genes fulfilling only LME criteria. Data underlying this scheme can be found in Supplementary Data 8.

ANOVA on ranks; Fig. 8e–g), suggesting a constitutively active current. A nonsense variant (c.965G > A, p.(Trp322*)) was identified by exome sequencing in a patient with high-functioning ASD. An intragenic deletion of the first coding exon of *TRPC5*, encoding conserved ankyrin repeats, was previously reported in a patient with ASD who had a family history compatible with X-linked inheritance[52]. In addition, three de novo variants in *TRPC5* (p.(Pro667Thr), p.(Arg71Gln), p.(Trp225*)) had been identified in patients with intellectual disability and/or autism disorders in the literature[21,53,54]. The two nonsense variants and the intragenic deletions theoretically lead to a loss-of-function of the TRPC5 channel, possibly by degradation of the corresponding mRNA by the nonsense-mediated decay (NMD) system of the cell. However, this hypothesis could not be tested due to the unavailability of patients' material.

Altogether, these results strongly suggest that pathogenic variants altering the functions of *CDK16* and *TRPC5* lead to novel X-linked disorders featuring ID and ASD.

## Discussion

Male and female individuals tolerate pathogenic variants on chrX in different manners. Variants in X-linked genes have to be interpreted taking this complexity into account. We aimed at providing an inventory of disorder genes on chrX and predict genes that remain to be associated with human disease, assuming that they share similar characteristics. We first used a threshold approach to list genes similarly constrained during evolution, which are the most likely to lead to disease when altered by genetic variants. This revealed 127 genes not yet known in human pathology sharing at least one constraint metric

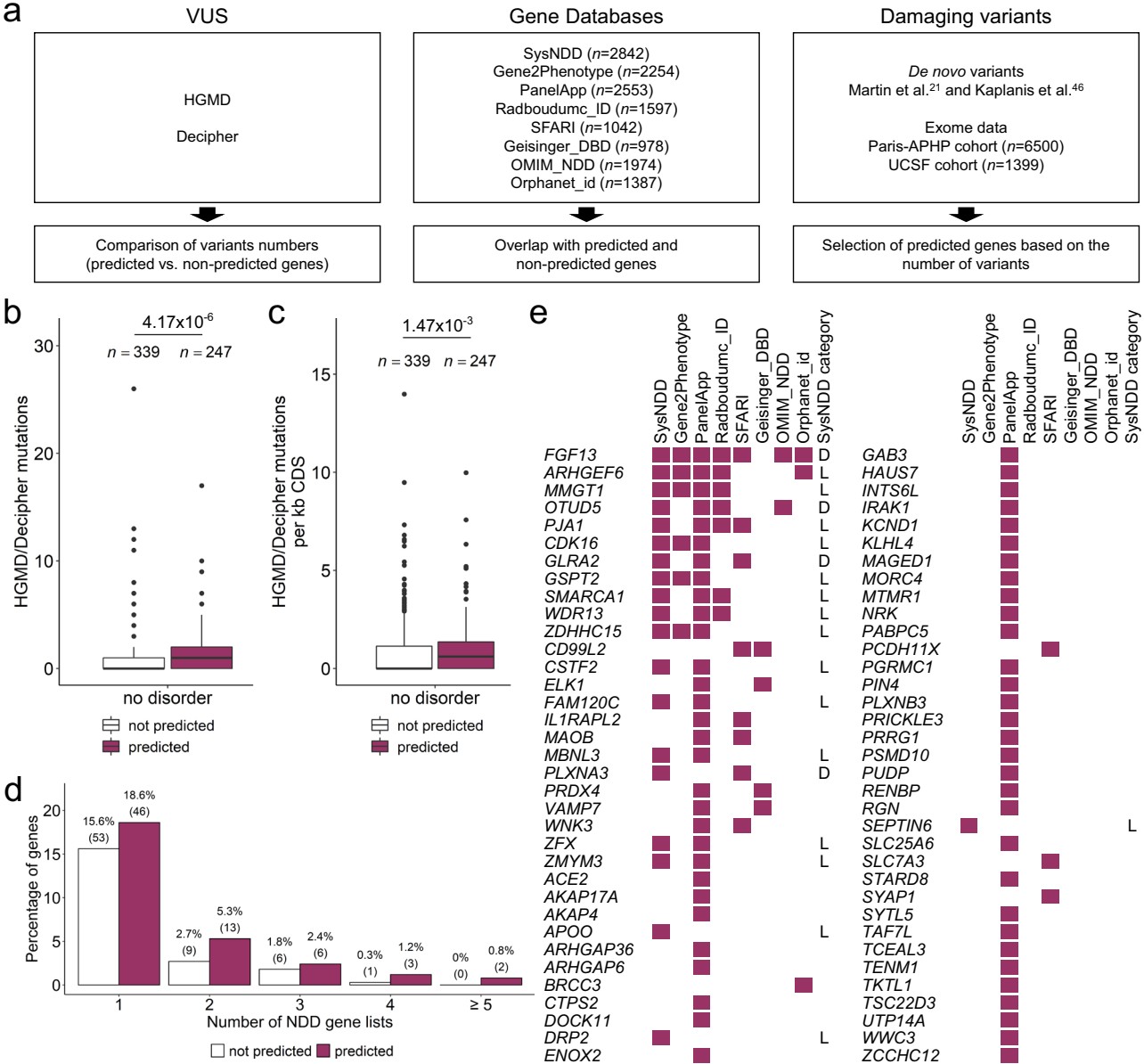

Fig. 6 | External supporting evidence for the predicted disorder-associated genes. a Sources of data reinforcing predicted genes as putative disorder-associated included variant or expert curated NDD gene databases, literature and exome data examination from two additional cohorts of patients with developmental disorders. Known point mutations in no-disorder genes. Boxplot showing the number of known mutations reported in HGMD and DECIPHER (b) or their value normalized by coding-sequence (CDS) length (c) according to their predicted status. Box plot elements are defined as follows: center line: median; box limits: upper and lower quartiles; whiskers: 1.5× interquartile range; points: outliers. d Percentage of each class of no-disorder genes present in the expert curated NDD gene databases. Number of genes are shown in brackets. b–d White, non-predicted no-disorder genes; purple, predicted no-disorder genes. Source data are provided as a Source Data file. e ML-predicted genes present in expert curated NDD gene databases. Genes in SysNDD are classified as D (definite NDD gene) or L (limited evidence as NDD gene). Data underlying this scheme can be found in Supplementary Data 8.

with known disorder genes and 35 sharing at least two. To avoid bias and limitations linked to this approach, we used a hypothesis-free machine learning approach to differentiate genes associated with brain disorders from genes where homozygous damaging variants are tolerated (dispensable genes). The ML-classifiers predicted 247 genes as putative disorder genes, including most of the genes uncovered by the threshold approach.

Our predictions are supported by a higher number of point variants reported in DECIPHER and HGMD in predicted versus non-predicted genes, which strengthens the probability that variants reported contribute to the patients' phenotypes and prompts the prioritization of these genes in further genetic analyses. We notably highlight 13 genes in which several possibly pathogenic variants in

functional domains have been identified in patients, likely constituting novel genes associated with neurodevelopmental disorders. Furthermore, a subset of predicted genes (e.g. OTUD5, FGF13, GLRA2, PLXNA3 GPKOW, and WNK3) have already been associated with human diseases, although these associations had no OMIM Gene-Phenotype Relationship entries when we started our study. Furthermore, we provide additional evidence that variants in two predicted genes, CDK16 and TRPC5, likely cause X-linked disorders by gathering genetic and clinical data of unrelated families with damaging variants in these genes. We expect that pathogenic variants in most predicted genes will be associated with neurological features, and more particularly intellectual disability, language impairment and/or seizures, as this is the case for the eight genes discussed above. However, it is possible that

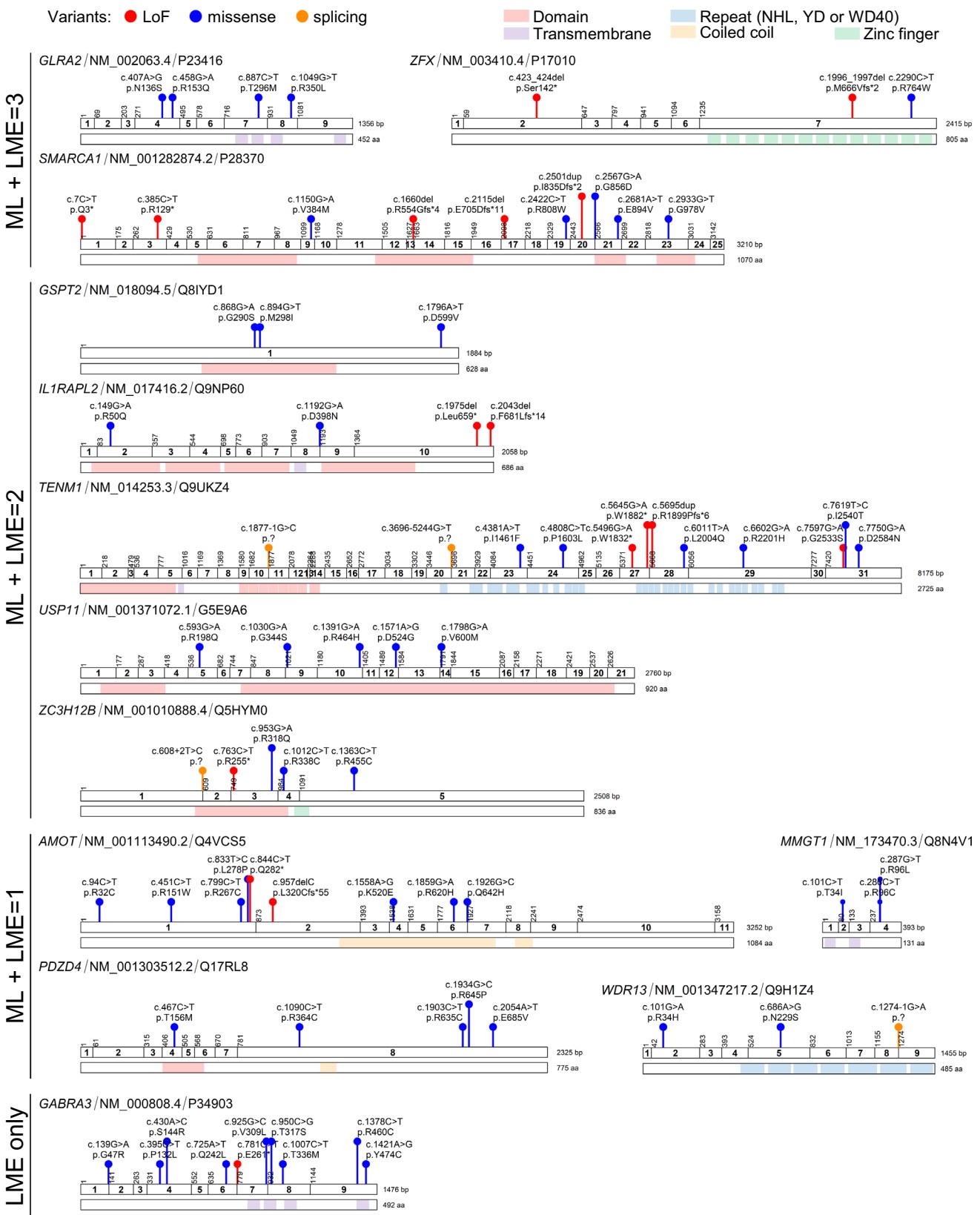

**Fig. 7 | Damaging variants in selected predicted disorder-associated genes.** Schematic representation of the coding exons, protein domains (when present) and available damaging variants (truncating or CADD ≥ 25) for each selected predicted gene. Types of variants are shown in different colors: lof-of-function (LoF, red), missense (blue), splicing (orange). Protein functional domains are shown: domains (light red), transmembrane segments (purple), NHL, YD or WD40 repeats (light blue), coiled-coils (yellow), zinc-fingers (green). HGVS cDNA and HGVS protein descriptions are shown. The corresponding RefSeq identifier of the MANE Select transcript and Uniprot identifier are shown for each gene. Details of variants displayed in this figure appear in Supplementary Data 10. The schemes were generated with ggplot2.

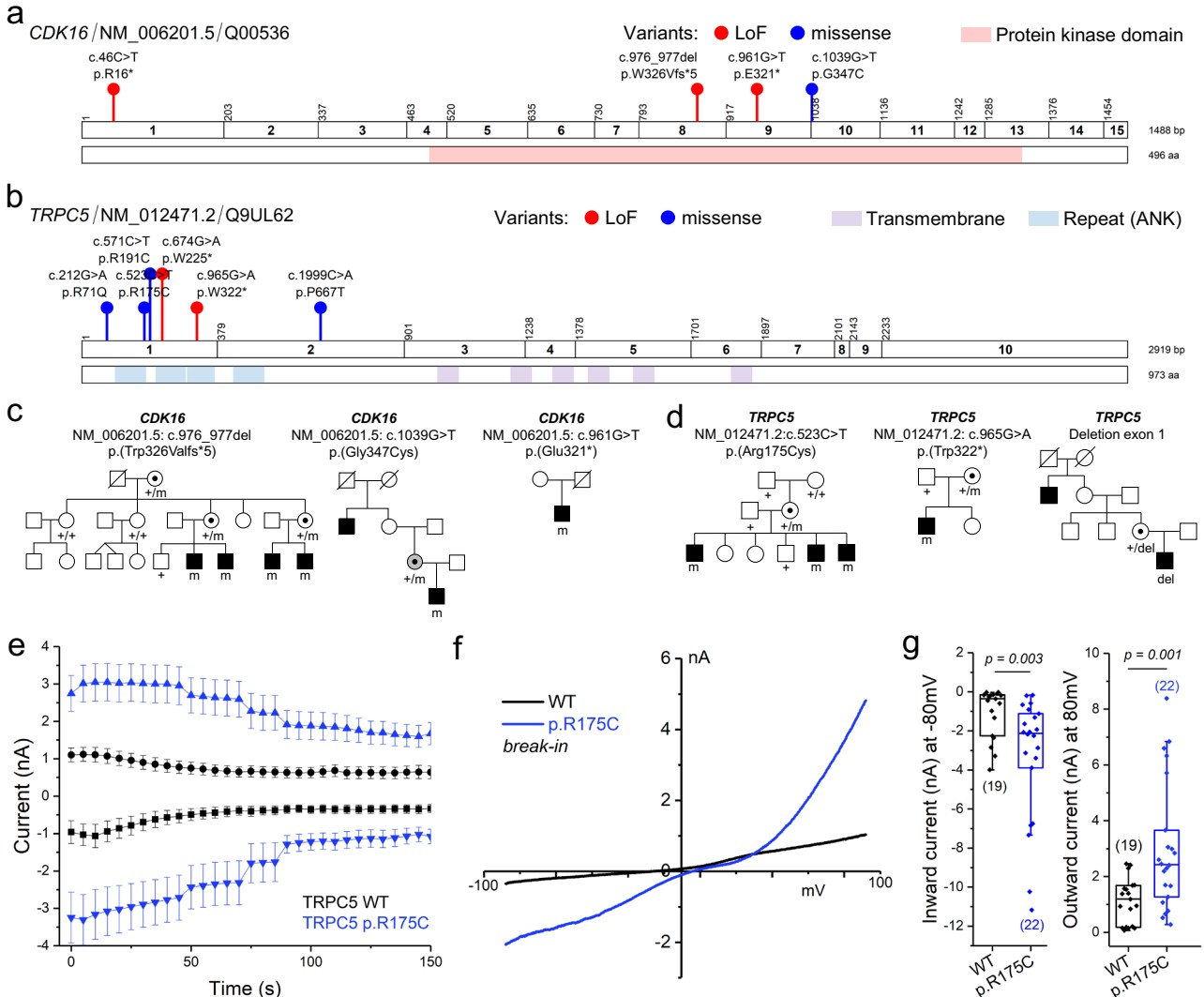

**Fig. 8 | Validation of selected predicted disorder-associated genes: *CDK16* and *TRPC5*.** Schematic representation of the coding exons, protein domains, and variants of *CDK16* (**a**) and *TRPC5* (**b**). Variant types: loss-of-function (LoF, red), missense (blue). Protein functional domains are shown: protein kinase domain (light red), transmembrane segments (purple), ANK repeats (light blue). Details of variants displayed in this figure appear in Supplementary Data 11. The schemes were generated with ggplot2. **c** Pedigrees of the three families with *CDK16* variants. The pedigree of family with c.976_977del, p.(Trp326Valfs*5) was adapted from Hu et al.[18]. The two remaining families are unpublished. **d** Pedigrees of the three families with *TRPC5* variants. The family with an intragenic exon deletion was reported in Mignon-Ravix et al.[52]. The two other families are novel. **e**–**g**, Functional characterization of *TRPC5* p.R175C by whole-cell patch-clamp recordings showing that the mutation renders the mutant channel constitutively opened. **e** Time course of inward and outward current amplitudes measured at +80 mV and −80 mV in HEK293 cells transiently expressing WT (black) and mutant TRPC5 (blue) in presence of 100 nM free Ca$^{2+}$ in the pipette. Values are reported as mean ± SEM (WT: $n = 18$ (0 s), $n = 19$ (5–70 s), $n = 17$ (75–100 s), $n = 15$ (105 s), $n = 14$ (110 s), $n = 13$ (115–135 s), $n = 12$ (140 s), $n = 11$ (145–150 s); mutant TRPC5: $n = 22$ (0–45 s), $n = 21$ (50–70 s), $n = 18$ (75–85 s), $n = 17$ (90–125 s), $n = 16$ (130–145 s), $n = 15$ (150 s)). Recordings started few seconds after the rupture. **f** Representative whole-cell current-voltage (I-V) relationships of WT and mutant TRPC5 channels current obtained shortly after break-in (≤10 s) with 100 nM free Ca$^{2+}$ in the pipette. **g** Boxplot of WT and mutant whole-cell current at −80 mV and +80 mV after break-in. Box plot elements are defined as follows: center line: median; box limits: upper and lower quartiles; whiskers: 1.5× interquartile range. The number of independent recordings appears in brackets. Statistical analyses were performed using Kruskal-Wallis one-way ANOVA on ranks Source data are provided as a Source Data file.

some genes will be associated with disorders manifesting in tissues or organs other than the brain, since the ML-classifiers also predicted genes known to be associated with non-neurological disorders. This result is not surprising given that most genes related to monogenic conditions will share common features regarding constraints and conservation.

Altogether, our gene-focused approach suggests that less than half of genes associated with human pathology on chrX are known so far and many more remain to be characterized clinically, which challenges the recent estimation that X-chromosome has been saturated for disease genes[21]. The differences in conclusions may come from recruitment bias in the DDD study, with inclusion of families with

X-linked inheritance in more specific studies, as well as from the patient-driven approach itself, which only allows detecting what is statistically significant in a sample, no matter how large this sample is. It is indeed possible or even likely that phenotypes associated with genes we predict are ultra-rare, lethal in utero, difficult to recognize or assess clinically, or associated with atypical modes of X-linked inheritance, making their identification difficult using classical genetic approaches.

Focusing on known genes associated with monogenic conditions in the first part of this study, we observed a higher proportion of genes associated with disorders on chrX compared to autosomes. This finding suggests that family-driven approaches have been more

efficient in identifying disorder-associated genes on this chromosome. Nevertheless, we also observed an enrichment of genes associated with specific neurological features such as ID, seizures, and language delay on chrX. This enrichment, which is based on genes associated with disorders and not all genes on the chromosomes, is therefore independent of the proportion of disorder genes identified on each chromosome. The specific association of X-linked genes with cognitive functions has largely been conveyed in the literature but, to our knowledge, this study is the first demonstrating the statistical significance of this finding using a systematic unbiased analysis. The reason why so many genes important for cognition and language are on a sex chromosome is fascinating but remains so far mysterious[55].

Our study also indicates that disorder genes on chrX are depleted in highly similar paralogues (>95% identity) compared to PMT/no-disorder genes, a finding that remained significant for predicted genes compared to non-predicted ones. The consequence of the existence of paralogues for disorders is probably different depending on the expression of these paralogues and the ability of the gene product to compensate the function of the original genes. Interestingly, 20% of genes on chrX have paralogues with >95% identity. This list includes copies or retrocopies on autosomes, a functional redundancy that has been attributed at least partly to the transcriptional silencing of chromosome X during spermatogenesis[56,57]. Although many gene copies or retrocopies are specifically expressed in male germ cells, others are still broadly expressed and could therefore buffer genetic variants in the X-linked paralogous genes, as shown for *UPF3B* and *UPF3A*[58]. The existence of paralogues is also linked in some ways to the location in PAR regions and escape to XCI. Indeed, genes in PAR have paralogues on chrY and escape inactivation[59]. Accordingly, 33 of the 242 predicted genes outside PAR also show some degree of escape from X inactivation (Supplementary Data 2). Interestingly, genes escaping XCI outside PAR, including *IQSEC2* and *KDM5C*, may lead to disorders manifesting in both sexes[10,12]. Our study indicates that genes in PAR and genes with close paralogues are less likely to be associated with a disorder, suggesting that paralogues can compensate pathogenic variants in some X-linked genes and raising the possibility of digenism or oligogenism in genes with redundant functions.

Our study focused on coding genes and did not include genes encoding long non-coding RNA (lncRNA) or other RNA classes, which are however very abundant on chrX. Only a few non-coding genes have been associated with disease so far and we believe that constraints and pathological mechanisms applying to non-coding genes are likely different from those of coding genes. In this respect, our study only predicts genes associated with disorders when affected by usual mutation types. Therefore, our findings do not exclude that non-predicted genes could lead to disease when associated with unusual mechanisms, such as gain of a new function, dominant-negative impact on another gene, and ectopic expression of a gene in the wrong tissue or at an abnormal time during development.

In conclusion, our study provides new insights into the complexity of X-linked disorders and indicates that alternative approaches not initially based on patient cohorts are effective to reveal gene-disease associations. We provide a list of genes that are likely to be associated with human disorders. Further studies are required to delineate these disorders clinically and determine whether males and/or females harboring variants in these genes are affected.

## Methods
### Protein-coding genes
Annotations of genes in all chromosomes were downloaded from the HUGO Gene Nomenclature Committee (HGNC) database in November 2020 focusing on protein-coding genes with approved status and present in the reference assembly[60]. *MED14OS* was excluded from chrX-focused analysis due to its encoded protein being curated in Uniprot as "Product of a dubious CDS prediction". Information concerning genes in PARs was also retrieved from HGNC.

### Gene-phenotype relationships
Gene associations with disorders and/or traits were retrieved from data files provided by Online Mendelian Inheritance in Man (OMIM) in November 2020. Information concerning the Clinical Synopsis (CS) of phenotypes was obtained through OMIM Search. OMIM morbid genes were annotated for (1) the availability of Clinical Synopsis data for any of the associated phenotypes, (2) the existence of Clinical Synopsis neurologic features in any of the associated phenotypes and (3) the presence of terms related to intellectual disability, seizures, impaired language development, impaired motor development, spasticity and ataxia among the Clinical Synopsis neurologic features. For the latter and due to the free-text nature of the Clinical Synopsis data, synonymous terms and sentences of the above-mentioned clinical features were used and their complete lists are shown in Supplementary Data 3. Genes were divided into three groups based on the existence and type of their gene-phenotype relationships: (1) "confirmed" when associated with at least one confirmed monogenic disorder; (2) "PMT" for genes either with provisional gene-phenotype relationship (P), or associated with susceptibility factors to multifactorial disorders (M) or with traits (T), which are labelled in the OMIM Gene Map with a question mark, braces or brackets, respectively; (3) "no disorder" for genes showing no phenotype associations. For each chromosome, we calculated the fraction of protein-coding genes that are CS-genes and the fraction of CS-genes that contain non-specific or specific neurologic features in the associated Clinical Synopsis data.

### Gene intolerance to variation
Metrics related to intolerance of chrX genes to genetic variation (loss-of-function (LoF), missense and synonymous) were retrieved from the Genome Aggregation Database (gnomAD) v2.1.1[23] after conversion of gene identifiers to HGNC approved symbols. For genes with available data for multiple transcripts, the metrics of the one with lowest LoF observed/expected upper bound fraction (LOEUF) were kept.

### Gene expression data
Median gene-level transcripts per million (TPM) for 54 tissues from the v8 release were downloaded from the GTEx portal[24]. The robust tissue specificity measure tau[61,62] was calculated for chrX genes, briefly: the median expression of each gene was aggregated for the different brain regions into two values: (i) the median of the values for the two cerebellar regions and (ii) the median of the values for the other brain tissues; $\log_2(TPM+1)$ median expression values were calculated for all tissues; and tau (τ) was calculated for each gene as: $\tau = \sum_{i=1}^{N}(1-x_i)/(N-1)$, where $N$ is the number of tissues and $x_i$ is the expression profile component normalized by the maximal component value[22]. Genes with tau below 0.6 are broadly expressed, while tau higher than 0.6 indicates genes expressed in a restricted number of tissues.

Additional expression data was downloaded from the BrainSpan Atlas of the Developing Human Brain[25]. The Developmental Transcriptome Dataset contains gene summarized RPKM (Reads Per Kilobase of transcript, per Million mapped reads) normalized expression values generated across 13 developmental stages in 8-16 brain structures from a total of 42 individuals of both sexes. After conversion of RPKM to TPM and gene identifiers to HGNC approved symbols, we restricted to genes on chrX and calculated the mean TPM according to age groups, ignoring sex and tissue origin: (1) Pre-natal; (2) Post-natal; (2a) Post-natal 1: after birth until four years-old (inclusive); (2b) Post-natal 2: older than four years old.

For the machine learning, the expression data was stratified by sex: (1) we calculated the mean and variance expression of each gene

for females and males independently, for brain, cerebellar tissues and nerve, aggregating in one metavalue the data for the two cerebellar tissues and in another the data for the other brain regions; (2) tau values were also calculated independently for females and males; (3) Brainspan mean and variance TPM for each age group was calculated separately for females and males.

## Canonical transcript selection

The criteria for canonical transcript selection prioritized: (i) MANE (Matched Annotation between NCBI and EBI) Select transcripts, which were independently identified by both Ensembl and NCBI as the most biologically relevant; and (ii) APPRIS-annotated transcripts[63]. TSS, MANE, and APPRIS annotations of transcripts were retrieved from Ensembl Genes 102 via BioMart[64]. For most genes (ca. 80%), the canonical transcripts belong to the MANE Select category. The longest transcript with APPRIS annotation was selected for ca. 20% of genes. In one case, the only existing transcript was kept.

## Promoter CpG density

CpG density was calculated for the 4 kb region surrounding the transcription start site (TSS ± 2 kb) of the canonical transcripts of chrX protein-coding genes following previous publications[22,65]. Promoter sequences were downloaded through UCSC Table Browser[66]. The CpG density, defined as the observed-to-expected CpG ratio, was calculated as follows: number of CpG dinucleotides/(number of cytosines × number of guanines) × N, where N is the total number of nucleotides in the sequence being analyzed.

## Exon and promoter conservation across species

Nucleotide conservation across 100 vertebrate species was calculated using the phastCons score obtained with the phastCons100-way.UCSC.hg38 R package[67], and represents the probability that a given nucleotide is conserved (range 0 to 1). Exon-conservation score was calculated for each protein-coding gene as the average phastCons of all nucleotides belonging to the gene coding sequence. Coding coordinates for the canonical transcripts were retrieved from Ensembl Genes 102[64] using biomaRt R package[68,69]. Promoter-conservation score for each protein-coding gene was calculated as the average phastCons score of all nucleotides 4 kb around the TSS of the canonical transcript.

## Distance to centromere and telomeres

The coordinates from centromeres and telomeres were downloaded through UCSC Table Browser. We calculated the distance from the TSS of the canonical transcripts to the centromere and to the telomere in the corresponding chromosomal arm.

## Annotation of encoded proteins

Data referring to function, subcellular location, subunit structure, and gene ontology terms for chrX encoded proteins were retrieved from Uniprot[70].

## Paralogues

All paralogues from protein-coding genes were retrieve from Ensembl Genes 102[64] using biomaRt R package[68,69]. The 90th, 95th, 98th and 99th, and 100th percentiles were calculated for (i) the percentage of paralogous sequence matching the query sequence (target %) and (ii) the percentage of query sequence matching the paralogue sequence (query %). Only paralogues with both metrics above the 95th percentile were considered as close paralogues.

## X-chromosome inactivation

The information on genes escaping X-chromosome inactivation (XCI) was obtained from multiple publications[4,71–76]. After converting gene identifiers from all studies into HGNC approved symbols, genes were

divided into seven categories based on the agreement between the various studies: (1) high-confidence escapee and (2) high confidence non-escapee, when almost all studies agreed on one status; (3) low confidence escapee and (4) low confidence non-escapee, whenever some studies disagreed but a higher number of studies reported one status; (5) variable escapee, when most studies stated variable escape; (6) discordant, when similar number of studies agreed on both status; (7) not available, when there was not enough data to have reliable evidence of XCI status.

## Features shared by disorder-causing genes

To uncover no-disorder genes exhibiting similar characteristics to known disorder-causing genes, we considered metrics showing enrichment of confirmed-disorder genes at one of the extremes of the distribution with marked difference between confirmed-disorder and no-disorder genes. Then, we applied a threshold approach to categorize genes showing values within the deciles enriched for confirmed disorder-causing genes for each of the considered metrics. The minimum number of criteria showing enrichment for confirmed disorder-causing genes was used as the minimum number of criteria required to consider a PMT or no-disorder gene as having similar features to disorder-causing genes.

## Pre-classification of genes for the machine learning

Genes were pre-classified based on (1) the type of associations with disorders and/or traits (confirmed, PMT, no-disorder), (2) the association with a brain disorder and (3) the tolerance to loss-of-function (LoF) variants (Fig. 4b; Supplementary Data 1). The list of confidently LoF-tolerant genes based on the gnomAD dataset was retrieved from data files from Karczewski et al. (2020) and consists of genes with at least one homozygous LoF variants[23]. Genes showing no homozygous LoF variants were considered LoF-intolerant.

## Machine Learning

Information regarding the data fed into the machine learning classifiers is in Supplementary Data 5 and 6. The data comprise 83 features collected from 19,154 genes that included 35 gene constraints from gnomAD[23], two nucleotide conservation metrics, 35 features related to expression data stratified according to sex (19 based on GTEx multitissue data from adults[24] and 16 based on Brainspan data) focusing on brain development[25], four gene structure attributes, two were aspects of the relative gene position on the chromosome, and five were related to paralogues. We applied nested cross-validation with outer 10-fold and inner 5-fold cross-validation to 25 different classification/regression machine learning models provided by scikit-learn[26]. Different combinations for a model and corresponding model-specific space of hyperparameters were tested either by grid search or by random search as fallback if the number of combinations exceeds 100. The performance of each model/hyperparameter combination was measured by the Matthews correlation coefficient (MCC). Genes of the classes "Cbi" (class B, value 1.0) and "NDt" (class D, value 0.0) were used as training data and in a preprocessing step all features were normalised to the range [0.0;1.0] (Supplementary Data 1 and 6). If applicable, we recalibrated the estimated probabilities for each classifier by applying a CalibratedClassifierCV. The Supplementary Data 7 shows the characteristics of all 25 classifiers, their different parameter spaces, estimated MCC mean and standard deviation by the nested cross-validation, and the final detected best parameter settings. By definition of the MCC, the base value is 0.0. As expected, we see this for the DummyClassifier with a slight deviation due to the limited cardinality of the data and the statistical imbalance during the training/test split in the cross-validation process.

We took the top five classifiers ranked by their mean MCC and built a common prediction by averaging their probabilities. Afterwards we estimated the threshold t for a given false discovery rate d by first

defining functions $D(t)$ and $B(t)$ returning all genes of class $D$ respectively $B$ with score $<t$. We then iteratively decreased $t$ until $D(t)/(B(t) + D(t)) < d$. We considered genes with probability > 0.5 in the five classifiers, and also set a FDR < 0.05 on the mean probability of the ML-classifiers (probability > 0.82). The feature importance was estimated by random permutation[77]. We performed Kernel SHapley Additive exPlanations (Kernel SHAP)[78] to explain the prediction on selected genes. 100 samples from the training data were randomly selected as background and the model was reevaluated 50 times for each prediction. Shap values were visualized by force plots[79].

## Known mutations

We automatically retrieved point variants affecting the coding sequence (*i.e.* missense, nonsense, small deletions, small insertions, and small indels) of predicted and non-predicted genes listed in the Human Gene Mutation Database (HGMD) Professional 2020.3 (Qiagen) as disease-causing (DM) or variants of unknown significance (DM?) excluding splicing variants, regulatory variants, gross deletions, gross insertions, complex rearrangements and repeat variations. HGMD professional is a manually curated database that gathers all variants published in genetic studies. Some studies are based on a few families and include functional tests of the variants while others are based on detection of variants in particular patient cohorts according to specific criteria (e.g. de novo occurrence in trio exome analysis). Generally, variants reported in these studies are rare (with a MAF used for the cut-off differing from one study to the other) and comprise variants usually described as of high/moderate effect in NGS pipelines. The list can occasionally include other variant types (such as synonymous variants shown to have an impact on splicing). For specific genes (*KIF4A, NLGN3, NRK, ATP11C, GPKOW*, genes included in Figs. 7 and 8), variants were manually curated. Point variants, excluding variants listed as benign or likely benign, were in parallel, retrieved from DECIPHER[80]. DECIPHER is a database containing data from 43,129 patients, including notably patients studied as part of the Decipher Developmental Disorders (DDD) consortium. This database is used by the clinical community to share and compare phenotypic and genotypic data. Variants in predicted and non-predicted genes were listed from "matching patient variants" and "matching DDD research variants".

## Developmental disorders gene databases

chrX genes were annotated for their classification in SysNDD (https://sysndd.dbmr.unibe.ch/)[27] in June 2022 and the provided compared curations from Gene2Phenotype, PanelApp, Radboudumc, SFARI, Geisingel_DBD, OMIM_NDD, and Orphanet_id, which list genes associated with intellectual disability or developmental disorders.

## Identification of damaging variants in predicted genes

De novo variants in predicted genes were retrieved from Kaplanis et al.[46] and Martin et al.[21]. In addition, we examined variants in predicted genes identified in unsolved cases with developmental disorders from two different cohorts: a cohort from Hôpital Pitié-Salpêtrière (APHP, Paris, France), which mainly includes patients with intellectual disability or neurological disorders (6500 individuals from 2346 families) and a cohort of patients with dysgenesis of the corpus callosum, associated with learning or intellectual disabilities from University of California San Francisco (UCSF; 1399 individuals from 463 families). Informed consent of the legal representatives and appropriate approval of an ethics committee, according to the French and American laws, have been obtained. Variant data and clinical information were shared anonymously. We retrieved CADD PHRED scores[81,82] for all variants and only damaging variants (i.e. variants leading to a premature termination codon or missense variants with CADD PHRED scores ≥ 25) in ML-predicted genes meeting at least one LME criterion were kept for further analysis. In parallel, we also used Genematcher[48] to identify additional families with variants in selected genes.

## Functional validation of TRPC5 missense variant

The human NM_012471.2 TRPC5 isoform cloned in fusion to GFP at its C-terminus in a pCMV6-AC backbone was obtained from Origene (RG213238) and the p.Arg175Cys variant was introduced by site-directed mutagenesis. HEK293 cells were transiently transfected with a plasmid expressing either WT or mutant channels. Currents were recorded from green fluorescent cells using the whole-cell configuration of the patch-clamp technique 16 to 24 h after transfection at room temperature. Voltage-clamp recordings were performed using an Axopatch 200B amplifier and a Digidata 1440 Analogue/Digital interface (Axon Instruments, Molecular Devices). Data acquisition was performed with the Axon Clampex 11 Software (Axon™ pCLAMP™ Software Suite). Data were low-pass filtered at 2 kHz, digitized at 10 kHz and analysed offline using Axon Clampfit 11. Currents were recorded during a 500 ms voltage ramp from −100 mV to 100 mV applied from a holding potential of −80 mV every 5 s. Series resistance was not compensated, and no leak subtraction was performed. Data were not corrected offline for voltage error and liquid junction potential. The pipette solution contained (in mM): NaCl 8, Cs-methanesulfonate 120, $MgCl_2$ 1, $CaCl_2$ 3.1, EGTA 10, HEPES 10, pH adjusted to 7.3 with CsOH. The extracellular solution contained (in mM): NaCl 140, $MgCl_2$ 1, $CaCl_2$ 2, HEPES 10, glucose 10, pH adjusted to 7.2 with NaOH. Immediate currents were recorded upon break-in (using patch pipettes that contained 100 nM free $Ca^{2+}$). The mutant currents are readily distinguished even in the absence of agonist stimulation, indeed the current-voltage relationship (IV) was similar to that described in the literature, showing inward and outward ('double') rectification, giving something that is roughly 'Nshaped'. Englerin A (100 nM) was applied to the cell expressing WT or mutant TRPC5 without immediate current upon break-in to confirm its functional expression.

## Statistics

Fisher's tests (two-sided) were performed for each chromosome to determine associations between: (i) genes being present in the specific chromosome and genes having an associated phenotype with Clinical Synopsis data (CS-genes); (ii) CS-genes being in the specific chromosome and CS-genes having non-specific or specific neurologic features described in the Clinical Synopsis data. For chrX, Fisher's tests (two-sided) were performed to compute the enrichment/depletion of (i) confirmed disorder-associated genes in each decile of the distribution of continuous variables and (ii) non-predicted genes in each decile of the distribution of relevant features used as input for the machine learning classifiers. Odds ratios were $log_2$ transformed and indicate enrichment or depletion of genes, for positive or negative values, respectively ($10^{-4}$ was added prior to transformation, whenever necessary to deal with log of 0 issue). $P$-values were adjusted for multiple comparisons using Bonferroni correction. For gene prediction, Fisher's tests (two-sided) were performed to calculate the association between genes being confirmed-disorder genes and the number of predictors. Mann-Whitney U test (two-sided) followed by Bonferroni correction for multiple testing was used to assess (i) the coding-sequence length difference between groups of genes and (ii) the difference between HGMD/Decipher mutations between predicted and non-predicted genes. Correlation between the CDS length and the number of HGMD or Decipher mutations was calculated using Pearson's (two-sided) method (stat_cor function from the ggpubr R package). For the functional validation of *TRPC5* missense variant we used Kruskal-Wallis one-way ANOVA on ranks.

## Reporting summary

Further information on research design is available in the Nature Research Reporting Summary linked to this article.

## Data availability

Source data are provided with this paper. The data generated in this study are provided in the Supplementary Information, Supplementary Data or Source Data files. We used publicly available data from HUGO Gene Nomenclature Committee (HGNC) Database (https://www.genenames.org/)[60], Online Mendelian Inheritance in Man (OMIM) (https://www.omim.org/)[83], gnomAD v2.1.1 (https://gnomad.broadinstitute.org/)[23], GTEx Portal (https://gtexportal.org/home/)[24], the Developmental Transcriptome dataset from BrainSpan (https://www.brainspan.org/)[25], Ensembl (https://www.ensembl.org/index.html)[64], UCSC Table Browser (https://genome.ucsc.edu/cgi-bin/hgTables)[66], Uniprot (https://www.uniprot.org/)[70], DECIPHER (https://www.dechipergenomics.org/)[80], SysNDD database (https://sysndd.dbmr.unibe.ch/)[27], and CADD - Combined Annotation Dependent Depletion (https://cadd.gs.washington.edu/)[81,82]. In addition, two published datasets from Martin et al.[21] and Kaplanis et al.[46] and variants from the Human Gene Mutation Database (HGMD) professional database (commercially distributed by Qiagen) were used. Source data are provided with this paper.

## Code availability

The code used for the machine learning part and corresponding results are available on Github using the following link: https://doi.org/10.5281/zenodo.7031826 (ref. 84).

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

## Acknowledgements

The authors thank the patients with *CDK16* and *TRPC5* pathogenic variants and their family for their participation in this study. We kindly thank Mrs Céline Cuny for Sanger sequencing analysis of the patient with *TRPC5* missense variant. Electrophysiological experiments were carried out at the electrophysiology core facility of ICM funded from the program "Investissements d'avenir" ANR-10-IAIHU-06. We thank Universitätsklinikum Essen, University Duisburg-Essen, the Deutsche Forschungsgemeinschaft (DFG), the Tom-Wahlig-Stiftung (TWS), and the Deutsche Stiftung Neurologie (DSN) for their financial support to the research studies conducted by the authors. E.L. and F.K. are associated with the FOR 2488 project (DFG, Project number 287074911). Sequencing and analysis (Paris cohort) were supported by Assistance Publique des Hôpitaux de Paris (APHP). Sequencing and analysis (UCSF cohort) were provided by the Broad Institute of MIT and Harvard Center for Mendelian Genomics (Broad CMG) and was funded by the National Human Genome Research Institute (grant number: R01NS058721 to E.H.S.). This study makes use of data generated by the DECIPHER community. A full list of centres who contributed to the generation of the data is available from https://deciphergenomics.org/about/stats and via email from contact@deciphergenomics.org. Funding for the DECIPHER project was provided by Wellcome. E.A., E.H.S., C.M., D.H., and C.De. are members of the IRC[5] consortium.

## Author contributions

E.L. and C.De. conceived and supervised the study. E.L. and C.S. performed the computations. E.L., I.P., C.Da., A.R., T.K., S.K., N.D., and C.E. carried out the experiments. A.K., B.G., E.S., C.N., J.P., B-D-B., L.V., A.P.A.S., E.K.V., J.A.J.V., F.J.K., F.T.M-T., M.S., P.S., S.G.M.F., E.A., E.H.S., F.E., J.B., B.K., C.M., D.H., J-L.M., J.G., V.M.K., B.H., and A.P. contributed analysis, data and/or critically revised the manuscript for intellectual content. E.L. and C.De. drafted the manuscript. All authors reviewed and approved the manuscript.

## Funding

## Competing interests

The authors declare no competing interests.

## Ethics/informed consent

Informed consents have been obtained for each patient/participant included in this study. Patient/participant/samples were anonymized for the genetic study at each participating center. Participants receive no compensation for inclusion in the genetic study. Genetic and clinical data shared in the context in this study cannot be used to identify individuals. Participating centers and data collection sites had study protocols approved by the local Institutional Review Boards (IRB) including INSERM (RBM C12-06) and Ethik Kommission der Medizinischen Fakultät der Universität Duisburg-Essen. Researchers and clinicians from participating centers contributing either data or intellectual input were involved at all stages of the study from design, implementation, drafting, and revising the manuscript, and are coauthors of the article.

## Additional information

**Correspondence and requests** for materials should be addressed to Christel Depienne.

[1]Institute of Human Genetics, University Hospital Essen, University Duisburg-Essen, Essen, Germany. [2]Institut du Cerveau et de la Moelle épinière (ICM), Sorbonne Université, UMR S 1127, Inserm U1127, CNRS UMR 7225, F-75013 Paris, France. [3]Unité de Génétique Moléculaire, IGMA, Hôpitaux Universitaire de Strasbourg, Strasbourg, France. [4]Service de Génétique Médicale, IGMA, Hôpitaux Universitaires de Strasbourg, Strasbourg, France. [5]Institut de Génétique et de Biologie Moléculaire et Cellulaire, Illkirch 67400, France. [6]Centre National de la Recherche Scientifique, UMR7104, Illkirch 67400, France. [7]Institut National de la Santé et de la Recherche Médicale, U964, Illkirch 67400, France. [8]Université de Strasbourg, Illkirch 67400, France. [9]Centre de Génétique Humaine, CHU Besançon, Besançon, France. [10]INSERM UMR1231, Equipe Génétique des Anomalies du Développement, Université de Bourgogne-Franche-Comté, Dijon, France. [11]Centre de génétique chromosomique, Hôpital Saint-Vincent de Paul, Lille, France. [12]Aix-Marseille University, INSERM, MMG, UMR-S 1251, Faculté de médecine, Marseille, France. [13]Département de Génétique Médicale, APHM, Hôpital d'Enfants de La Timone, Marseille, France. [14]Department of Human Genetics, Radboud University Medical Center, 6500 HB Nijmegen, The Netherlands. [15]Department of Clinical Genetics, Maastricht University Medical Center+, Maastricht, The Netherlands. [16]Cardiovascular Research Institute (CARIM), Departments of Cardiology, Maastricht University Medical Center, Maastricht, The Netherlands. [17]Unité Fonctionnelle Innovation en Diagnostic génomique des maladies rares, CHU Dijon Bourgogne, Dijon, France. [18]Department of Neurosciences, Rehabilitation, Ophthalmology, Genetics, Maternal and Child Health, University of Genoa, 16132 Genoa, Italy. [19]Pediatric Neurology and Muscular Diseases Unit, IRCCS Istituto Giannina Gaslini, 16147 Genoa, Italy. [20]Department of Genetics and Cell Biology, Faculty of Health Medicine Life Sciences, Maastricht University Medical Center+, Maastricht University, Maastricht, The Netherlands. [21]Department of Neurology, University of

California, San Francisco, San Francisco, CA, USA. [22]Institute of Human Genetics and Weill Institute for Neurosciences, University of California, San Francisco, San Francisco, CA, USA. [23]UF de Génomique du Développement, Département de Génétique, Groupe Hospitalier Pitié-Salpêtrière, APHP-Sorbonne Université, Paris, France. [24]APHP, Sorbonne Université, Département de Génétique, Centre de Référence Déficiences Intellectuelles de Causes Rares, Groupe Hospitalier Pitié-Salpêtrière and Hôpital Trousseau, Paris, France. [25]School of Medicine, The University of Adelaide, Adelaide 5005 SA, Australia. [26]Robinson Research Institute, The University of Adelaide, Adelaide, SA 5006, Australia. [27]South Australian Health and Medical Research Institute, The University of Adelaide, Adelaide 5005 SA, Australia. [28]Research Group Development and Disease, Max Planck Institute for Molecular Genetics, Berlin, Germany. [29]These authors contributed equally: Elsa Leitão, Christopher Schröder. ✉e-mail: christel.depienne@uni-due.de

