## [Peer Review File · Nature Communications]

Systematic analysis and prediction of genes associated with monogenic disorders on human chromosome XREVIEWER COMMENTS

Reviewer #1 (Remarks to the Author):

The manuscript by Leitão et al., perform a systematic analysis of human chrX. They report damaging variants in TRPC5 gene affecting male sibling's different families. To find out if the mutation affects the functionality of the channel, the authors recorded the activity of the WT and mutant channel expressed in HEK cells. The results show a constitutive activity, with the distinctive current-voltage relationship of TRPC5 channels, in cells expressing the mutant channel; the experiments a well-performed but there are few number of experimental issues needed addressing.

Although there are some controversy in the field, it is described that TRPC5 may be activated by pressure, e.g. pipette pressure. In many cases, TRP channels exhibit basal constitutive activity on the plasma membrane. In fact according to fig. 6g, some cells expressing the WT channel also have constitutive activity. Would it be possible that the observed difference in the averaged current amplitude found between the WT and mutant is due to the two outlier cells that express the mutant channel?

It is mention that Englerin A was applied to cells expressing WT and mutant TRPC5 without immediate current upon break-in to confirm its functional expression, which is the proportion of cells that are activated upon break-in both cases? Does Englerin A potentiate currents in the mutant channel?

Minor comments:

Do the other variants found in patients with intellectual disabilities affect the activity of the channel?

Please, indicate the species of the TRPC5 cDNA. Does the mutant also express GFP?

Indicate the number of cells averaged in figure 6e and whether they are the same as in figure 6g.

Reviewer #2 (Remarks to the Author):

This manuscript characterizes known disease-associated genes on the X-chromosome, and uses these characteristics to try and prioritize other X-chromosome genes by their likelihood to be involved in disease. The work is well motivated, and makes good use of combining information across existing databases. The use of machine learning to search for features predictive of disease relevance of interesting, and there are efforts made to validate the resulting classifications against independent data.

It wasn't precisely clear how the main output of this work - a set of X-linked genes which have not yet been associated with disease, but which are computational predicted to have disease relevance if mutated - would be formally incorporated into future research. I can imagine that this prioritization would be of interest to people wanted to select X-linked genes for basic science characterization - although there is a lot of that yet to be done for the genes are are unambiguously disease relevant. I was less clear how any potential prioritization in clinical genetic protocols would work however - given that existing selective constraint metrics provide the main workhorse for estimated pathogenicity in loss of function variants. I imagine a third use might be candidate gene association studies, but I think the field is looking to other study designs as gold-standard evidence of association.

Despite these uncertainties, I think the systematic annotation provided here is useful - although the anticipated breadth of interest and impact from this may make the work better-suited for a more specialized journal that Nature Communications.

A few specific questions/comments:

Lines 130-132: "Furthermore, genes on chrX were significantly more frequently associated with neurological phenotypes than genes on autosomes". Does this hold if excluding genes that only lead

to phenotypes in males? Also, the methods describe BF correction was applied, but it is not clear what was considered a “family” of tests for this correction (i.e. in Fig 1c-n: across chromosomes per - phenotype, or additionally across phenotypes?) could the authors please clarify

CNN - I don't have expertise in neural networks, and it would be important for part of the peer review team to cover this

Validation of putative novel genes - I appreciate that this manuscript is focused on the X-chromosome, but the novel autosomal genes (presumably much larger than the novel X-linked genes) would have provided a good setting to further benchmark the prediction against independent annotations (as is so far only done for the X). This larger sample size could yield evidence of validity in prediction of novel genes that could be generalized to some extent to the X

Lines 260-261: “Interestingly, 19 out of the 55 missense variants reported in these 13 genes are localized in known functional protein domains (Fig. 5; Supplementary Table 11).” Does this observation deviate significantly from null expectations?

Focus on CDK16 and TRPC5 - it would be good to have more information regarding the basis for picking these two genes in particular, the cohorts from which the clinical cases were derived, and the grounds for using the specific functional assay mentioned for TRPC5

Reviewer #3 (Remarks to the Author):

Identification of novel disease-associated gene on the human X chromosome is challenging because balance of gene dosage by X chromosome inactivation predicts multiple forms of inheritance in these disorders. In this interesting study, Leitao and colleagues described a systematic analysis of human X chromosome genes and strategies to predict genes that remain to be associated with human X-linked diseases. The study reported a higher proportion of disorder-associated genes and an enrichment of genes involved in cognition, language, and seizure on X chromosome as compared to autosomes. The authors first applied a threshold approach to predict association of “non-disorder genes” with human X-linked diseases using an evolutionary constraint metric established with known disorder genes. Because X chromosome is enriched with genes associated with developmental brain disorders, the authors next utilized machine learning to train a neural network to distinguish disease-associated from dispensable genes to identify X chromosome genes with high probability of disease-association. The authors then provided evidence of an excess of variants in these predicted disease genes using several existing genome and disease databases in supporting their approaches and conducted genetic and/or functional studies of damaging variants in CDK16 and TRPC5 in patients with intellectual disability or autism spectrum disorders. This study result predicts that large-scale gene-disease associations could be used to prioritize X-linked genes and pathogenic variants.

This study described a valuable approach to predict novel human X-chromosome genes-associated with diseases and provided multiple lines of solid evidences supporting this approach. The study is well-designed and presented. The work is significant and original. The resulting dataset would be very useful to researchers in this field. Clarification and modifications are needed to strengthen the manuscript.

Comments to the Authors

1. Clarifications are needed. “Dispensable genes” are used frequently. Is this description the same as “no-disorder genes”? “Disorder-associated genes” are also used frequently. It is not clear if only genes causing X-linked Mendelian disorders are included as this study largely aims to predict if any X-linked genes associated with increased risks to diseases are included here.

2. The authors stated that “a higher proportion of disorder-associated genes and an enrichment of genes involved in cognition, language, and seizures on chrX compared to autosomes.”

Comments. These data are nicely presented and supported in Figure 1 based on current OMIM entries for X-linked monogenic disorders and X chromosome genome. Similar comments were also made previously by others. However, most X-linked disorders were identified by distinct phenotypes segregated with unique X-linked inheritance in proband families. As a result, X-linked disorders in general are relatively easy to recognize and characterize as compared to autosomal disorders. Due to the nature of X chromosome, many X-linked disease loci are among the earliest genes being mapped and documented in OMIM as distinct disease entities. This raises a competing hypothesis that disease phenotypes associated with genes on the X chromosome are better and systematically characterized as compared to disorders associated with genes on the autosomes. Thanks in part to the systematic efforts to study genetics of X-linked ID disorders by several groups over recent decades, these research works result in much better understanding of X-linked ID genes compared with genes associated with other X-linked disorders.

The “threshold approach” uses a set of standard evolutionary constraint metrics established with known disorder genes to predict disease association of X-linked non-disorder genes.

Comments: While this is a valuable approach, it is generally predicted that disorder-associated genes show lower LOEUF values and higher misZ scores as noted in the manuscript. Are there any reasons that disorder-associated genes on autosomes and X chromosome behave differently using these prediction criteria?

The “machine learning approach” trains a neural network to distinguish known disorder genes associated with neurological features and dispensable genes in order to “predict remaining disorder genes in systematic and unbiased fashion” on the X chromosome.

Comments. Since X chromosome is “enriched” for genes causing developmental brain disorders of cognition and language development, it is understandable to use neural network in the proposed machine learning to predict novel genes associated with these groups of disorders. However, it should also be assumed that this neural network-based machine learning approach is not suitable to predict disease-associated genes involving other disease categories on the X-chromosome.

The authors stated that “we provide evidence of an excess of variants in predicted genes in existing databases.”

Comments. It is not immediately clear in the manuscript what types of variants are used for this comparison. It is assumed that these variants are rare with certain MAF cutoff and are predicted to be functionally deleterious. If so, a clear description of these criteria is needed.

For validation of putative novel disorder genes, the authors “retrieve the number of single nucleotide variants and indels of unknown significance reported in each gene and compared the number of variants in predicted versus non-predicted gene categories”. The study identified “235 NN-predicted disorder genes that are enriched in point variants compared to non-predicted genes suggesting that this excess is due to pathogenic variants”.

Comments. It is not immediately clear on the nature of these “single nucleotide variants” in the manuscript. Do these “indels of unknown significance” predict truncating gene products that implicate a deleterious function? Please clarify.

Reviewer #4 (Remarks to the Author):

This work describes efforts to characterize the disease potential of all genes present on chrX in humans. Overall, the manuscript is very well written and enjoyable to read, particularly for someone who is not an expert in chrX biology. However, the machine learning analysis seems somewhat preliminary in nature and some modest revisions and additional experiments could significantly

strengthen the story. I hope that these comments help the authors produce a stronger manuscript and I look forward to seeing another version after my concerns are addressed.

1. I think that the story regarding training a machine learning model needs to be fleshed out more. I understand the motivation is that there are many genes on chrX whose relationship with disease is unclear. However, because genes can be modified in several ways, e.g. changes to the protein sequence, changes to the regulatory logic, alternate splicing, etc., that the question "is this gene involved with a neurological disease" is somewhat crude and doesn't shed clues as to the mechanism. Further, after the model is trained and an initial evaluation is performed the model seems to be largely discarded. The final section involves a thorough analysis of two genes that are predicted positives without an explanation of why these two were selected from the hundreds of positives. It seems like these two could have been chosen without the need for machine learning at all.

2. It was unclear to me what exactly the prediction problem being solved was when reading "Prediction of novel disorder..." other than genes are being predicted to be either known disorder genes or dispensable. When a machine learning method is presented it is critical to clearly write out what all of the input features are (at least a high-level description with a complete description in the methods), the labels, and where the examples came from. Given that small details in how machine learning is done, particularly with neural networks, can mean the difference between correct and incorrect analysis, being clear on the details in the text is crucial. Having the input features be in one supplemental table, the training and architecture details be in the methods, and the evaluation details in another supplemental figure makes it challenging to assess the correctness of the evaluation.

3. I think that a more comprehensive and rigorous analysis of the machine learning model is necessary. Right now, it's unclear to me that the model is performing well because I generally don't find taking a subset of positive predictions from the model and showing that they match up with some external data source to be particularly convincing. Specifically, presenting evaluations that emphasize recall, e.g. "X of Y known disease genes were predicted", don't speak to the precision, which is arguably more important when prioritizing future experimental efforts. Instead, I think that the paper should include a more traditional analysis of performance alongside simple baselines to evaluate how well the model does. The paper already includes a 10-fold cross-validation on the autosomes to calculate the FDR values. Calculating FDR values seems unnecessary to me, but using this cross-validation to evaluate model performance on autosome and presenting those performance values in the main text would be very useful.

4. Another challenge with assessing the performance of the model is that no baselines are presented. When showing results from the 10-fold cross-validation (or whatever splits the authors choose), performance should also be given for simpler models to justify the added complexity of a neural network, e.g. just the average label (DummyClassifier in sklearn), a logistic regression model, etc. I also usually compare the performance of using each feature individually and report the top features just to get a sense for which features are predictive and get more non-ML baselines.

5. Details are given on the model architecture but it is unclear how the authors chose those particular hyperparameters. It is common to use hyperparameter search to find the best model on some sort of validation set. If the authors did not do hyperparameter selection, they should consider using at least randomized grid search to search a portion of the space. If they did hyperparameter selection, they should clearly describe what procedure they used. Importantly, the best model should be determined on a held out validation set and not the result of the overall cross-validation (see nested cross-validation).

6. I think that it would be scientifically insightful to use feature attributions methods to, for each gene, identify why the model thought that the gene was disease-associated. Methods like DeepLIFT/DeepSHAP, integrated gradients, ISM, etc are commonly used in the field of genomics to interpret deep learning models and highlight the motifs underpinning their predictions. Here, the trained neural networks using sklearn won't play too well with the commonly used implementations of DeepLIFT/DeepSHAP, but KernelSHAP (in the SHAP repository) could be used instead. Such an

analysis here could potentially be invaluable for scientists that may be skeptical of looking at computational predictions alone, and could also be useful in explaining the rules that the model learned.

Responses to reviewer comments

We would like to thank the reviewers for their positive assessment and for providing constructive and helpful comments that led to improve our manuscript.

Reviewer #1:

The manuscript by Leitão et al., perform a systematic analysis of human chrX. They report damaging variants in TRPC5 gene affecting male sibling's different families. To find out if the mutation affects the functionality of the channel, the authors recorded the activity of the WT and mutant channel expressed in HEK cells. The results show a constitutive activity, with the distinctive current-voltage relationship of TRPC5 channels, in cells expressing the mutant channel; the experiments a well-performed but there are few number of experimental issues needed addressing.

Although there are some controversy in the field, it is described that TRPC5 may be activated by pressure, e.g. pipette pressure. In many cases, TRP channels exhibit basal constitutive activity on the plasma membrane. In fact according to fig. 6g, some cells expressing the WT channel also have constitutive activity. Would it be possible that the observed difference in the averaged current amplitude found between the WT and mutant is due to the two outlier cells that express the mutant channel?

We thank the reviewer for these comments. The electrophysiological experiments included in our manuscript aimed to demonstrate that the *TRPC5* missense variant identified in patients has an impact on the encoded channel function. We are aware that the heterologous expression of mutant in cell lines, which are classical to demonstrate pathogenicity of variants in ion channels remain limited to basic in vitro assays and do not permit to address the pathophysiological mechanisms related to this mutation, which are of great interest but clearly out of the scope of this manuscript. In particular, the possibility that the mutation alters the ability of the channel to activate upon mechanical stress has not been tested. The same pressure "treatment" (including pressure of the pipette on the cell, pressure induced during whole-cell access to the membrane, same intracellular and extracellular solution were used for both groups to avoid difference in osmolarity) has been applied during patch-clamp recordings of cells expressing WT and mutant channels. Thus, the results we obtain for the mutant channel are unlikely to come from differences in pipette pressure.

We have checked that the difference observed between WT and mutant conditions remains significant even removing the two outliers from our previous dataset:

$p = 0.007$ for the inward current, (Kruskal-Wallis one-way ANOVA on ranks with Dunn's multiple comparisons test)

$p = 0.0335$ for the outward current (one-way ANOVA with Dunnett's multiple comparisons test)
 $n = 13$ for WT and $n = 12$ for mutant

In addition, we now provide additional recordings strengthening our observations. Here are the results of the statistical analysis conducted without the outliers for the combined dataset:

$p = 0.002$ for the inward current, (Kruskal-Wallis one-way ANOVA on ranks with Dunn's multiple comparisons test),

$p = 0.009$ for the outward current (Kruskal-Wallis one-way ANOVA on ranks with Dunn's multiple comparisons test)

$n = 19$ for WT and $n = 20$ for mutant

The full dataset, highlighting outliers, has now been added as Supplementary Table 15.

It is mention that Englerin A was applied to cells expressing WT and mutant TRPC5 without immediate current upon break-in to confirm its functional expression, which is the proportion of cells that are activated upon break-in both cases? Does Englerin A potentiate currents in the mutant channel?

26% of cells expressing WT and 9 % of cells expressing mutant TRPC5 did not show immediate current with the characteristic shape upon breaking but were nicely responding to Englerin A. Only one cell was

excluded from the analysis in the WT group as the cell did not show an immediate current and no Englerin A-induced current.

Englerin A was not automatically tested in cells that are activated upon break-in with immediate current but when we did we had difficulty maintaining a long recording upon application of Englerin A and most of the time the cell was rapidly lost. The effect of Englerin A on the mutant channel is out of the scope of this work and could be studied in a follow up study on TRPC5.

Minor comments:

Do the other variants found in patients with intellectual disabilities affect the activity of the channel?

Other variants identified in patients with ASD are nonsense or large intragenic deletions that are both predicted to lead to a loss-of-function of the TRPC5 channel. The likely consequence of these variants is the degradation of the mRNA by the nonsense-mediated decay (NMD) system, which would lead to an absence of protein synthesis but we did not have patient material to test this hypothesis. We have now added a sentence in the text (page 16, lines 355-358) to make this point clear.

Please, indicate the species of the TRPC5 cDNA. Does the mutant also express GFP?

Indicate the number of cells averaged in figure 6e and whether they are the same as in figure 6g.

The plasmid expresses human *TRPC5* (isoform NM_012471.2) fused to GFP at its C-terminus (from Origene). This has now been added in the results (page 16, line 342) and in the method (pages 29-30, lines 678-681). The mutant plasmid only differs from the WT plasmid at the site of the variant.

Indicate the number of cells averaged in figure 6e and whether they are the same as in figure 6g.

The number of cells averaged in Figure 8e is the same as in Figure 8g for the first 10 time points but differs thereafter. Indeed, the recording of the cell expressing the TRPC5 mutant did not last long, and we decided to keep only the cell being recorded for at least 50 seconds. Thus, the number of cells averaged in Figure 8e are the same for the first time point but depends thereafter on the time from which each cell was recorded without being lost. The legend of Figure 8e has been implemented as such: n = 11-19 for WT and n = 16-22 for mutant.

Reviewer #2:

This manuscript characterizes known disease-associated genes on the X-chromosome, and uses these characteristics to try and prioritize other X-chromosome genes by their likelihood to be involved in disease. The work is well motivated, and makes good use of combining information across existing databases. The use of machine learning to search for features predictive of disease relevance of interesting, and there are efforts made to validate the resulting classifications against independent data.

1-It wasn't precisely clear how the main output of this work - a set of X-linked genes which have not yet been associated with disease, but which are computational predicted to have disease relevance if mutated - would be formally incorporated into future research. I can imagine that this prioritization would be of interest to people wanted to select X-linked genes for basic science characterization - although there is a lot of that yet to be done for the genes are unambiguously disease relevant. I was less clear how any potential prioritization in clinical genetic protocols would work however - given that existing selective constraint metrics provide the main workhorse for estimated pathogenicity in loss of function variants. I imagine a third use might be candidate gene association studies, but I think the field is looking to other study designs as gold-standard evidence of association.

Despite these uncertainties, I think the systematic annotation provided here is useful - although the anticipated breadth of interest and impact from this may make the work better-suited for a more specialized journal than Nature Communications.

We thank the reviewer for these comments. We believe that the predictions we provide in our manuscript can be used in many different ways, depending on the interests of the reader (basic science, clinical genetics, inheritance of complex traits, etc.). The most immediate application is indeed to prioritize

variants in predicted genes that would be identified in patients with a still unsolved phenotype for further genetic/functional analysis. Our work also highlights the need to think differently when confronted to variants in X-linked genes as the tolerance of these variants strongly depends on the sex of the individual and X inactivation processes in females. We hope that our work will incite other groups to look at variants on chromosome X differently and also take paralogous genes, location in pseudo-autosomal regions and gene constraints into account when studying variants located in a gene on chromosome X. We agree with this reviewer that our work mainly applies to Mendelian/monogenic disorders, and the findings cannot directly be used for the study of complex traits. For this reason, we decided to modify the title of our manuscript (new title: Systematic analysis and prediction of genes associated with monogenic disorders on human chromosome X) to more accurately reflect its content. However, the compensation by paralogous genes may also be relevant for studies focusing on oligogenic interactions. Overall, we think that our study describes new insights about X-linked genes that are of broad interest to the genetic community and we feel confident that these results can be used in multiple creative ways in downstream studies.

A few specific questions/comments:

2-Lines 130-132: "Furthermore, genes on chrX were significantly more frequently associated with neurological phenotypes than genes on autosomes". Does this hold if excluding genes that only lead to phenotypes in males? Also, the methods describe BF correction was applied, but it is not clear what was considered a "family" of tests for this correction (I.e. in Fig 1c-n: across chromosomes per - phenotype, or additionally across phenotypes?) could the authors please clarify

We thank the reviewer for this suggestion. However, as most disorder genes on chr X are associated with phenotypes restricted to males so far (the recognition and validation of phenotypes in females are usually less straightforward due to problems interpreting variants in the context of X inactivation), this is not possible to test the hypothesis only for genes associated with phenotypes in females or both males and females as this comparison would be underpowered.

Regarding the Bonferroni correction, it was applied across both chromosomes and phenotypes (184 tests: 8 features * 23 chromosomes; together for features in Fig. 1c-n and Supplementary Fig. 1). We updated the legend for clarification (page 39, line 969-971).

3-CNN - I don't have expertise in neural networks, and it would be important for part of the peer review team to cover this

Please see our responses reviewer #4 regarding the part on machine learning.

4-Validation of putative novel genes - I appreciate that this manuscript is focused on the X-chromosome, but the novel autosomal genes (presumably much larger than the novel X-linked genes) would have provided a good setting to further benchmark the prediction against independent annotations (as is so far only done for the X). This larger sample size could yield evidence of validity in prediction of novel genes that could be generalized to some extent to the X

The machine learning methods developed in this manuscript have been designed to predict genes associated with monogenic disorders on chrX. In particular, as the expression of X-linked phenotypes mainly depends on the sex of the individual, we stratified expression data according to sex to use as input for the machine learning classifiers. For this reason, and because predictions on autosomes likely need to distinguish other classes of genes than those distinguished in this Ms, the predictions using our ML-approach performs less effectively for autosomes compared to chr X (see supplementary Figures 6 and 7). To predict with more accuracy genes associated with disorders on autosomes, it is more relevant to separate genes associated with dominant and recessive inheritance rather than separating according to sex. We plan to address this in an independent study.

5-Lines 260-261: "Interestingly, 19 out the 55 missense variants reported in these 13 genes are localized in known functional protein domains (Fig. 5; Supplementary Table 11)." Does this observation deviate significantly from null expectations?

In this sentence, we state that many of the reported variants were located in important functional domains without referring to an enrichment. Nevertheless, we performed a binomial test to test this hypothesis. For this test, we considered 55 trials, 19 successes, and a 24% theoretical probability of success (2683/11089; number of amino acids in domains divided by the number of amino acids in the 13 proteins). The p-value obtained (0.08279) does not permit to claim an enrichment of variants in functional domains.

6- Focus on CDK16 and TRPC5 - it would be good to have more information regarding the basis for picking these two genes in particular, the cohorts from which the clinical cases were derived, and the grounds for using the specific functional assay mentioned for TRPC5

The strategy we chose to confirm predicted genes is described in the results section. After looking for a global evidence of variant excess in available databases, we looked for variants in predicted genes in exome data of two large cohorts (n=6500+1399). In parallel, we used Matchmaker exchange (Genematcher) to identify additional patients with variants in *TRPC5* and *CDK16* as we identified at least one patient with a possibly pathogenic variant in these two genes in our cohorts. *TRPC5* and *CDK16* were among the genes showing the strongest genetic (X-linked segregation of the variants in at least one family) and clinical (overlap of clinical features in patients with different variants) evidence. We have now added a sentence to clarify why these two genes were chosen in the results (page 15, lines 325-327) and in the methods (page 29, lines 674-675).

Reviewer #3:

Identification of novel disease-associated gene on the human X chromosome is challenging because balance of gene dosage by X chromosome inactivation predicts multiple forms of inheritance in these disorders. In this interesting study, Leitao and colleagues described a systematic analysis of human X chromosome genes and strategies to predict genes that remain to be associated with human X-linked diseases. The study reported a higher proportion of disorder-associated genes and an enrichment of genes involved in cognition, language, and seizure on X chromosome as compared to autosomes. The authors first applied a threshold approach to predict association of "non-disorder genes" with human X-linked diseases using an evolutionary constraint metric established with known disorder genes. Because X chromosome is enriched with genes associated with developmental brain disorders, the authors next utilized machine learning to train a neural network to distinguish disease-associated from dispensable genes to identify X chromosome genes with high probability of disease-association. The authors then provided evidence of an excess of variants in these predicted disease genes using several existing genome and disease databases in supporting their approaches and conducted genetic and/or functional studies of damaging variants in *CDK16* and *TRPC5* in patients with intellectual disability or autism spectrum disorders. This study result predicts that large-scale gene-disease associations could be used to prioritize X-linked genes and pathogenic variants.

This study described a valuable approach to predict novel human X-chromosome genes-associated with diseases and provided multiple lines of solid evidences supporting this approach. The study is well-designed and presented. The work is significant and original. The resulting dataset would be very useful to researchers in this field. Clarification and modifications are needed to strengthen the manuscript.

Comments to the Authors

1. Clarifications are needed. "Dispensable genes" are used frequently. Is this description the same as "no-disorder genes"? "Disorder-associated genes" are also used frequently. It is not clear if only genes causing X-linked Mendelian disorders are included as this study largely aims to predict if any X-linked genes associated with increased risks to diseases are included here.

We thank the reviewer for pointing out terms needing clarification. We now provide a definition of "no-disorder" and "dispensable" genes (page 7, line 143 and page 10, line 203 of the results, and Supplementary Table 1). No-disorder genes are simply genes that had not yet been associated with a human disorder at the time of our study. Dispensable genes refer to genes that can be knocked out on both alleles i.e. tolerate LoF variants in the homozygous state without apparent consequence on the viability / health of the individual (as defined by Karczewski et al. 2020). No-disorder genes therefore include both

dispensable genes and genes that will be associated with disorder in future studies and our goal was to make the distinction between these two categories. See also description and number of genes in each category in Supplementary Table 1.

We believe that our study mainly applies to X-linked monogenic disorders (see also response to point 1 from reviewer #2). For this reason, we decided to modify the title of the revised manuscript accordingly: Systematic analysis and prediction of genes associated with monogenic disorders on human chromosome X

2. The authors stated that "a higher proportion of disorder-associated genes and an enrichment of genes involved in cognition, language, and seizures on chrX compared to autosomes."

Comments. These data are nicely presented and supported in Figure 1 based on current OMIM entries for X-linked monogenic disorders and X chromosome genome. Similar comments were also made previously by others. However, most X-linked disorders were identified by distinct phenotypes segregated with unique X-linked inheritance in proband families. As a result, X-linked disorders in general are relatively easy to recognize and characterize as compared to autosomal disorders. Due to the nature of X chromosome, many X-linked disease loci are among the earliest genes being mapped and documented in OMIM as distinct disease entities. This raises a competing hypothesis that disease phenotypes associated with genes on the X chromosome are better and systematically characterized as compared to disorders associated with genes on the autosomes. Thanks in part to the systematic efforts to study genetics of X-linked ID disorders by several groups over recent decades, these research works result in much better understanding of X-linked ID genes compared with genes associated with other X-linked disorders.

We thank the reviewer for this comment. The higher proportion of disorder-associated genes on chr X is very likely due to the early recognition of male-restricted phenotypes within families that allowed to better and more systematically identify the altered genes and corresponding phenotypes. This point is discussed on page 18 (lines 408-409) of the revised manuscript. However, the enrichment of genes associated with disorders that include intellectual disability, language deficit, and seizures, is calculated taking into account only genes associated with disorders and not all genes on the chromosomes. This means that there are more genes associated with these three phenotypes on chrX than on other chromosomes independently of the number of disorder genes identified on each chromosome. This result is unlikely the consequence of a better and more systematic identification of diseases on chr X and rather reflects that chr X is overall enriched in genes involved in cognition, language, and seizures. We rephrased the sentence pages 18-19 (lines 412-414) to reflect more clearly this aspect.

The "threshold approach" uses a set of standard evolutionary constraint metrics established with known disorder genes to predict disease association of X-linked non-disorder genes.

Comments: While this is a valuable approach, it is generally predicted that disorder-associated genes show lower LOEUF values and higher misZ scores as noted in the manuscript. Are there any reasons that disorder-associated genes on autosomes and X chromosome behave differently using these prediction criteria?

This is an interesting question. Contrary to genes on autosomes, where males and female tolerate variants equally, chromosome X has the particularity to be in a single copy in males and two copies in females, one of which is randomly inactivated. We believe that the gene constraints applying to gene on this chromosome are therefore probably different from those on autosomes and also differ between males and females. Because LOEUF and misZ are calculated not taking sex of the individuals into account, it is possible that these metrics have intrinsic limitations on chrX. Our study suggests that these metrics are still valuable but could be perhaps adapted to calculate other, sex-specific metrics. For example, truncating mutations in genes like *PCDH19*, and *EFNB1* are expected to be tolerated in hemizygous males and may be present in a male control population while they would be depleted (because disease-causing) in healthy heterozygous females. The opposite situation would be true for most other genes associated with X-linked disorders, some genes being depleted in both populations (with variants being lethal in males while

disease-causing in females). For this reason, we also chose to develop a machine learning classifier specifically for genes on chr X, separating data from males and females.

The "machine learning approach" trains a neural network to distinguish known disorder genes associated with neurological features and dispensable genes in order to "predict remaining disorder genes in systematic and unbiased fashion" on the X chromosome.

Comments. Since X chromosome is "enriched" for genes causing developmental brain disorders of cognition and language development, it is understandable to use neural network in the proposed machine learning to predict novel genes associated with these groups of disorders. However, it should also be assumed that this neural network-based machine learning approach is not suitable to predict disease-associated genes involving other disease categories on the X-chromosome.

We thank the reviewer for this valid comment. Indeed, the machine learning approach was designed to predict genes associated with neurological disorders, but we show that it can be used to predict genes associated with other disorders (out of 50 chromosome X genes associated with human disorders not affecting the brain, 41 (82%) are predicted by the neural network, $FDR < 0.05$; Fig. 4b). This result is not very surprising as some of the most important metrics to predict gene-disease association are gene constraints. The expression profile of the gene does not predict the clinical features (e.g. some genes expressed only in the liver such as *PAH* (phenylketonuria) can also be associated with neurological disease and many genes expressed ubiquitously have a phenotypic expression only in the brain). We have now added a few sentences in the discussion (pages 17-18, lines 387-393) to address this point.

The authors stated that "we provide evidence of an excess of variants in predicted genes in existing databases."

Comments. It is not immediately clear in the manuscript what types of variants are used for this comparison. It is assumed that these variants are rare with certain MAF cutoff and are predicted to be functionally deleterious. If so, a clear description of these criteria is needed.

For validation of putative novel disorder genes, the authors "retrieve the number of single nucleotide variants and indels of unknown significance reported in each gene and compared the number of variants in predicted versus non-predicted gene categories". The study identified "235 NN-predicted disorder genes that are enriched in point variants compared to non-predicted genes suggesting that this excess is due to pathogenic variants".

Comments. It is not immediately clear on the nature of these "single nucleotide variants" in the manuscript. Do these "indels of unknown significance" predict truncating gene products that implicate a deleterious function? Please clarify.

We retrieved automatically point variants affecting the coding sequence listed in HGMD professional as disease-causing (DM) or VUS (DM?) excluding splicing variants (i.e. missense, nonsense, small deletions, small insertions and small indels), regulatory variants, gross deletions, gross insertions, complex rearrangements and repeat variations. HGMD professional is a manually curated database that gathers all variants published in genetic studies. Some studies are based on a few families and include functional tests of the variants while others are based on detection of variants in particular patient cohorts according to specific criteria (e.g. de novo occurrence in trio exome analysis). Generally, variants reported in these studies are rare (with a MAF used for the cut-off differing from one study to the other) and comprise variants usually described as of high/moderate effect in NGS pipelines. The list can occasionally include other variant types (such as synonymous variants shown to have an impact on splicing). Although the methods used to report these variants vary from one study to the other, we do not expect that this variability would affect the difference in variant numbers we see for predicted versus non-predicted genes, and this approach should therefore be valid to demonstrate an excess of variants in one of the two groups. This is now added pages 28 (lines 635-654) of the methods.

Reviewer #4:

This work describes efforts to characterize the disease potential of all genes present on chrX in humans. Overall, the manuscript is very well written and enjoyable to read, particularly for someone who is not an expert in chrX biology. However, the machine learning analysis seems somewhat preliminary in nature and some modest revisions and additional experiments could significantly strengthen the story. I hope that these comments help the authors produce a stronger manuscript and I look forward to seeing another version after my concerns are addressed.

1. I think that the story regarding training a machine learning model needs to be fleshed out more. I understand the motivation is that there are many genes on chrX whose relationship with disease is unclear. However, because genes can be modified in several ways, e.g. changes to the protein sequence, changes to the regulatory logic, alternate splicing, etc., that the question "is this gene involved with a neurological disease" is somewhat crude and doesn't shed clues as to the mechanism. Further, after the model is trained and an initial evaluation is performed the model seems to be largely discarded. The final section involves a thorough analysis of two genes that are predicted positives without an explanation of why these two were selected from the hundreds of positives. It seems like these two could have been chosen without the need for machine learning at all.

We have now improved the machine learning approach as suggested by this reviewer: we have trained and compared 25 machine learning models (see also responses below). We have modified the respective sections in the Results (pages 10-12, lines 198-260) and the Methods (pages 26-28, lines 601-632). We considered the best five models/classifiers (including a similar neural network used in the previous version of the manuscript). Although the results differ slightly from one model to the other, there is an important overlap in predicted genes, with the final list being very similar to that obtained with the previous approach.

As pointed out by this reviewer, our study does not permit to address the mechanisms associated with these putative new X-linked disorders. It is very likely, however, that most will be "conventional" mechanisms *i.e.* mainly loss-of-function of the corresponding genes, which are by far, the most frequent consequence of pathogenic variants causing monogenic disorders, and perhaps, to a lower extent gain-of-function. As we point out in the discussion, our predictions cannot be extended to the discovery of other, more unusual mechanisms such as ectopic expression of a gene in a tissue or any gain of a novel function or toxicity of the gene product.

The validation part includes a global analysis of predicted versus non-predicted genes but as validating >100 genes is a huge work that will likely take several years of research in multiple institutions and needs data from patient cohorts that are not available to us. We decided to focus on genes for which we could gather genetic and functional evidence directly from exome data accessible to us. While trying to validate other genes highlighted in figure 7, we were in contact with groups already gathering evidence of genes not included in the manuscript. These studies will be published separately from this work. Please also see response to point 6 of reviewer #2.

2. It was unclear to me what exactly the prediction problem being solved was when reading "Prediction of novel disorder..." other than genes are being predicted to be either known disorder genes or dispensable. When a machine learning method is presented it is critical to clearly write out what all of the input features are (at least a high-level description with a complete description in the methods), the labels, and where the examples came from. Given that small details in how machine learning is done, particularly with neural networks, can mean the difference between correct and incorrect analysis, being clear on the details in the text is crucial. Having the input features be in one supplemental table, the training and architecture details be in the methods, and the evaluation details in another supplemental figure makes it challenging to assess the correctness of the evaluation.

The data used to train the machine learning classifiers previously appeared in a supplementary table. We agree with this reviewer that the nature of data being used was not immediately clear and we now provide a summarized description of the input features for machine learning in both the results and methods, with a

detailed description of each feature provided in Supplementary Table 8. We hope that this new format will allow the reader to directly evaluate the information used to train the machine learning classifiers.

3. I think that a more comprehensive and rigorous analysis of the machine learning model is necessary. Right now, it's unclear to me that the model is performing well because I generally don't find taking a subset of positive predictions from the model and showing that they match up with some external data source to be particularly convincing. Specifically, presenting evaluations that emphasize recall, e.g. "X of Y known disease genes were predicted", don't speak to the precision, which is arguably more important when prioritizing future experimental efforts. Instead, I think that the paper should include a more traditional analysis of performance alongside simple baselines to evaluate how well the model does. The paper already includes a 10-fold cross-validation on the autosomes to calculate the FDR values. Calculating FDR values seems unnecessary to me, but using this cross-validation to evaluate model performance on autosome and presenting those performance values in the main text would be very useful.

We thank the review for this suggestion. The revised manuscript now includes a comparison of 25 ML classifiers and evaluate their performance using both Matthews correlation coefficient (MCC) and nested cross validation. We calculated the precision sensitivity for each chromosome for these classifiers. Data for the best five classifiers are now displayed in Supplementary fig. 6. We also decided to keep the FDR rate below 5% for each tool and also for the combined prediction using the five best ML classifiers.

4. Another challenge with assessing the performance of the model is that no baselines are presented. When showing results from the 10-fold cross-validation (or whatever splits the authors choose), performance should also be given for simpler models to justify the added complexity of a neural network, e.g. just the average label (DummyClassifier in sklearn), a logistic regression model, etc. I also usually compare the performance of using each feature individually and report the top features just to get a sense for which features are predictive and get more non-ML baselines.

We thank the reviewer for the advice aiming at strengthening our machine learning analysis. In the previous manuscript version we developed a Multilayer Perceptroneural (MLP) neural network implemented in keras and performance was measured by the MCC, which is expected to have a baseline value of 0.0 when using a dummy classifier. Based on the suggestions, we have now performed training with a total of 25 classifiers using scikit-learn (which also includes MLP neural networks) and were able to identify three additional classifiers with slightly better performance (Supplementary Fig. 4). The DummyClassifier was one of the 25 and we were indeed able to demonstrate a value close to 0.0. We finally decided to make the prediction using the top 5 classifiers ranked by their MCC, instead of using only the neural network. Although the genes predicted to be disease-associated by the current and previous approaches are quite similar, we believe that this new method is considerably more robust. As suggested, we now also perform feature importance analysis using permutation and give a feature usage overview for the top 5 classifiers (Supplementary Fig. 5).

5. Details are given on the model architecture but it is unclear how the authors chose those particular hyperparameters. It is common to use hyperparameter search to find the best model on some sort of validation set. If the authors did not do hyperparameter selection, they should consider using at least randomized grid search to search a portion of the space. If they did hyperparameter selection, they should clearly describe what procedure they used. Importantly, the best model should be determined on a held out validation set and not the result of the overall cross-validation (see nested cross-validation).

Thanks to the reviewer, we now perform a nested cross validation of the 25 classifiers with a 10-fold outer cross-validation and a 5-fold inner cross validation. A parameter space of classifier specific hyperparameters is discovered by grid search or random search if the amount of parameter combinations exceeds 100. The performance is again measured by MCC. The space and detected best parameters for each model are given in Supplementary Table 10.

6. I think that it would be scientifically insightful to use feature attributions methods to, for each gene, identify why the model thought that the gene was disease-associated. Methods like DeepLIFT/DeepSHAP, integrated gradients, ISM, etc are commonly used in the field of genomics to interpret deep learning models and highlight the motifs underpinning their predictions. Here, the trained neural networks using

sklearn won't play too well with the commonly used implementations of DeepLIFT/DeepSHAP, but KernelSHAP (in the SHAP repository) could be used instead. Such an analysis here could potentially be invaluable for scientists that may be skeptical of looking at computational predictions alone, and could also be useful in explaining the rules that the model learned.

We agree that this is valuable information, and as suggested by the reviewer, we used KernelShap to gain this insight. We provide force plots based on the calculated shap values to visualize explanations for the gene predictions. Due to the focus of this work, we limited the analysis to genes on chromosome X (available on github) and only included plots for genes of interest in Supplementary Fig. 11.

REVIEWER COMMENTS

Reviewer #1 (Remarks to the Author):

I am satisfied with most of the authors' responses to my critiques however I would like to stress one point.

The authors indicate that the electrophysiological experiments included in the manuscript aimed to demonstrate that the TRPC5 missense variant identified in patients has a functional impact on the encoded channel function.

In my opinion, the functional characterisation of an ion channel should go beyond recording its basal electrical activity. I agree that studying the pathophysiological mechanisms of the TRPC5 channel is not the main aim of this study, but the experiment shown in figure 8 only indicates that there is a difference in the basal activity of the mutant channel, which could suggest a change in the channel function. Thus, the sentence "we investigated the functional impact of the p.(Arg175Cys) variant on the TRPC5 channel using whole-cell patch-clamp", should be rephrased because it sounds too ambitious for what is shown.

In reply to reviewers' comments, the authors indicate that 26% of cells expressing WT and 9 % of cells expressing mutant TRPC5 did not show immediate current with the characteristic shape upon breaking. In my opinion, this would also suggest that the mutant channel is constitutively more active.

Minor comment:

In the reviewers' report it is indicated that the statistical analysis is performed without the outliers, and the statistical significance is $p= 0.002$ and $p= 0.009$ for the input and output current respectively ($n= 19$ for WT and 20 for mutant). However in the figure, the n for the mutant is 22 , so it is not clear from figure 8G whether the statistical significance corresponds to the data showed in the figure or to the analysis without the outliers. Confirm the statistical significance of the data included in figure 8G.

Reviewer #4 (Remarks to the Author):

The authors have largely addressed my concerns. I believe that the machine learning procedure is now much more clearly described and the findings are more robust.

Response to reviewer's comments :

REVIEWERS' COMMENTS

Reviewer #1 (Remarks to the Author):

I am satisfied with most of the authors' responses to my critiques however I would like to stress one point. The authors indicate that the electrophysiological experiments included in the manuscript aimed to demonstrate that the TRPC5 missense variant identified in patients has a functional impact on the encoded channel function. In my opinion, the functional characterisation of an ion channel should go beyond recording its basal electrical activity. I agree that studying the pathophysiological mechanisms of the TRPC5 channel is not the main aim of this study, but the experiment shown in figure 8 only indicates that there is a difference in the basal activity of the mutant channel, which could suggest a change in the channel function. Thus, the sentence "we investigated the functional impact of the p.(Arg175Cys) variant on the TRPC5 channel using whole-cell patch-clamp", should be rephrased because it sounds too ambitious for what is shown.

Response: We modified the corresponding sentence as following: "We investigated the basal properties of TRPC5 p.(Arg175Cys) mutant channel using whole-cell patch-clamp. (page 16, line 345-346).

In reply to reviewers' comments, the authors indicate that 26% of cells expressing WT and 9 % of cells expressing mutant TRPC5 did not show immediate current with the characteristic shape upon breaking. In my opinion, this would also suggest that the mutant channel is constitutively more active.

Response: We thank the reviewer for this comment. This point could be addressed in a follow-up study on TRPC5.

Minor comment:

In the reviewers' report it is indicated that the statistical analysis is performed without the outliers, and the statistical significance is $p=0.002$ and $p=0.009$ for the input and output current respectively ($n=19$ for WT and $n=20$ for mutant). However in the figure, the n for the mutant is 22, so it is not clear from figure 8G whether the statistical significance corresponds to the data showed in the figure or to the analysis without the outliers. Confirm the statistical significance of the data included in figure 8G.

Response: In Figure 8g, we have only kept the statistical analysis performed using the whole data set (\$p=0.009\$ for outward current and \$p=0.002\$ for inward current; \$n=19\$ for WT, \$n=22\$ for mutant TRPC5). This information as well as the test used has now been added in the results section of the main manuscript (page 16, lines 348-350).

We believe that the observed outliers correspond to natural biological variations of recordings obtained for the mutant channel and not to artefacts that should be removed for the statistics. Due to a previous question regarding whether statistical significance could be reached omitting outliers, we performed the statistical analysis without outliers (\$n=19\$ for WT and \$n=20\$ for mutant) and confirmed that the \$p\$ values would still be significant. But we think that having the \$p\$ values obtained after removing the outliers is a possible source of confusion for the reader. All data underlying this figure are in the source data file and can be used to calculate any statistical tests. Furthermore, we agreed to publish the reviews and our responses to points raised by reviewers, and the information about \$p\$ values without outliers will therefore also be accessible.